# Functional characterization of a single nucleotide polymorphism associated with Alzheimer's disease in a hiPSC-based neuron model

Lindsay R. Stolzenburg[1], Sahar Esmaeeli[1], Ameya S. Kulkarni[1], Erin Murphy[1], Taekyung Kwon[2], Christina Preiss[2], Lamiaa Bahnassawy[3], Joshua D. Stender[1], Justine D. Manos[2], Peter Reinhardt[3], Fedik Rahimov[1], Jeffrey F. Waring[1], Cyril Y. Ramathal[1]*

1 AbbVie Inc., North Chicago, Illinois, United States of America, 2 AbbVie, Cambridge Research Center, Cambridge, Massachusetts, United States of America, 3 AbbVie Deutschland GmbH & Co. KG, Neuroscience Discovery, Knollstrasse, Ludwigshafen, Germany

* cyril.ramathal@abbvie.com

**Data Availability Statement:** All relevant data are within the paper and its Supporting information files. Additionally, all RNA sequencing files are

## Abstract

Neurodegenerative diseases encompass a group of debilitating conditions resulting from progressive nerve cell death. Of these, Alzheimer's disease (AD) occurs most frequently, but is currently incurable and has limited treatment success. Late onset AD, the most common form, is highly heritable but is caused by a combination of non-genetic risk factors and many low-effect genetic variants whose disease-causing mechanisms remain unclear. By mining the FinnGen study database of phenome-wide association studies, we identified a rare variant, rs148726219, enriched in the Finnish population that is associated with AD risk and dementia, and appears to have arisen on a common haplotype with older AD-associated variants such as rs429358. The rs148726219 variant lies in an overlapping intron of the FosB proto-oncogene (*FOSB*) and ERCC excision repair 1 (*ERCC1*) genes. To understand the impact of this SNP on disease phenotypes, we performed CRISPR/Cas9 editing in a human induced pluripotent stem cell (hiPSC) line to generate isogenic clones harboring heterozygous and homozygous alleles of rs148726219. hiPSC clones differentiated into induced excitatory neurons (iNs) did not exhibit detectable molecular or morphological variation in differentiation potential compared to isogenic controls. However, global transcriptome analysis showed differential regulation of nearby genes and upregulation of several biological pathways related to neuronal function, particularly synaptogenesis and calcium signaling, specifically in mature iNs harboring rs148726219 homozygous and heterozygous alleles. Functional differences in iN circuit maturation as measured by calcium imaging were observed across genotypes. Edited mature iNs also displayed downregulation of unfolded protein response and cell death pathways. This study implicates a phenotypic impact of rs148726219 in the context of mature neurons, consistent with its identification in late onset AD, and underscores a hiPSC-based experimental model to functionalize GWAS-identified variants.

available from the GEO database (accession https://www.ncbi.nlm.nih.gov/geo/query/acc.cgi?acc=GSE239367.

**Funding:** The author(s) received no specific funding for this work.

**Competing interests:** The authors have declared that no competing interests exist.

## Introduction

Alzheimer's Disease (AD) is an age-related neurodegenerative condition that affects more than 35 million people worldwide, with cases projected to increase as life expectancy continues to rise [1]. AD is characterized by progressive memory loss and cognitive decline and is pathologically marked by amyloid beta plaques and tau-based neurofibrillary tangles [2]. Genetically, AD etiology exists as two distinct forms: 1) early-onset familial AD (FAD), caused by rare mutations in high-impact genes usually inherited in an autosomal dominant fashion, and 2) late-onset AD (LOAD), the more common form. Heritability estimates for LOAD are between 58–79% [3], suggesting significant genetic contribution to disease development and/or progression. However, the genetic landscape of LOAD is complex, and likely arises from the combination of many low-effect single nucleotide polymorphisms (SNPs), which has historically hindered the identification and functionalization of genetic risk factors.

More recently, the discovery of novel variant associations with LOAD risk has exploded, thanks to successful efforts by large-cohort genome wide association studies (GWAS) [4, 5]. The strongest genetic risk factor identified consistently across studies is apolipoprotein E (*APOE*), whereby the ε4 allele contributes approximately 3-fold increased risk for LOAD in heterozygotes and ~10-fold increased risk in homozygotes, compared to the non-risk ε3/ε3 genotype [6]. Furthermore, GWAS efforts have also identified at least 37 additional risk loci for LOAD to date, mostly in non-coding regions of the genome [4, 7]. Still, mechanistic details of how these GWAS-identified variants contribute to disease development, including by the *APOE* risk alleles, remain grossly incomplete.

Despite intense research efforts, no cure for AD has been found, and treatment options for cognitive decline and disease progression are extremely limited [8]. Traditional preclinical models for drug discovery, such as rodents or immortalized cell lines, recapitulate some of the pathophysiological hallmarks of AD, although discoveries from these models have had limited success in being translated to the clinic, hindering the development of new therapeutics. The debut of human induced pluripotent stem cells (hiPSCs), re-programmed somatic cells capable of differentiating into most cell types in the body (including brain cells), has provided the field with a vastly improved method to model neurodegeneration *in vitro* [9]. Combined with recent advances in CRISPR-Cas9 gene-editing technology, which allows the engineering of distinct genetic variants into cells [10, 11], hiPSCs provide a unique tool for investigating the impact of LOAD risk loci in a disease-relevant cellular model.

This study describes the identification and hiPSC-based genetic modeling of a previously unstudied LOAD-associated SNP, rs148726219. Identified from the FinnGen biobank of phenome-wide association studies (PheWAS) [12], rs148726219 is typically a rare allele but is highly enriched in the Finnish population. Finland has the highest mortality rate from dementia and AD in Europe, where 1 in 3 people over 85 are predicted to die from the disease [13]. By studying this SNP in a non-Finnish genomic context, we uncovered specific SNP-based effects on disease mechanisms independent of other SNPs in linkage disequilibrium (LD) with rs148726219. We find that rs148726219 does not have any impact on differentiation of hiPSCs to induced neurons (iNs); however, transcriptome-wide profiling reveals genotype-dependent changes specifically in mature iNs in several biological pathways directly relevant to AD, particularly calcium signaling. Additional cell type-independent gene expression changes were also identified in CRISPR-edited clones throughout hiPSC-iN differentiation.

## Materials and methods

### hiPSC cell lines and gene editing

The BIONi010-C-13 line (donor biosample SAMEA103988285) with doxycycline (DOX)-inducible neurogenin 2 (*NGN2*) transgene was obtained and is available from the European Bank for Induced pluripotent Stem Cells (EBiSC, www.ebisc.org) [14]. The line is derived from a dermal fibroblast sample donated by an 18 year-old healthy male of African-American descent. The EBiSC Bank acknowledges SAMEA103988285 as the source of human induced pluripotent stem cell line BIONi010-C-13 which was generated with support from EFPIA companies and the European Union (IMI-JU).

CRISPR/Cas9-mediated SNP editing of BIONi010-C-13 was completed by the Genomic Engineering & iPSC Center (GEIC) at Washington University School of Medicine. To generate clones Heterozygous for rs148726219, the parental BIONi010-C-13 line was transfected with HiFi Cas9 v3 (IDT), gRNA with sequence GAGCTGTGGGTGGCTAGGGC, and ~60 bp single-stranded DNA oligos as homology arms flanking the PAM site (left HA: cccgggctctttctacccttaacactctggaaagcctgtgaaatgaaattattccacctcT; right HA: ctagccacccacagctctcctggtgctggtggcatcccccaaaacccactcccttcctac). An episomal vector expressing GFP was used for selection of transfected cells by FACS-based single-cell sorting. Clones were expanded and genotypes confirmed by next generation sequencing (NGS). The wild-type clone was generated in the same CRISPR experiment but was untargeted. To generate homozygous clones, heterozygous clones were re-targeted a second time.

Cytogenetic analysis was performed by the GEIC on all generated clones using STR profiling and G-banding of 20 metaphase cells. All cells exhibited normal male karyotypes (46, XY). Whole genome sequencing was performed to validate rs148726219 editing and evaluate off-target effects.

hiPSC clones were maintained in Essential 8 or Stem Flex media (Gibco) on Matrigel-coated plates (77.5 ug/ml, Corning) and passaged twice weekly as dissociated cells using Accutase (Gibco) and Revitacell supplement (Gibco).

### Differentiation of induced neurons (iNs)

Directed differentiation of hiPSCs to excitatory iNs was performed as previously described, with modifications [15, 16]. Briefly, hiPSCs were seeded on Matrigel-coated plates at ~25,000 cells/cm$^2$. The next day (day 0), culture media was replaced with N2/B27(-vitA)/DMEM/F12 media (Gibco) containing 2 ug/ml DOX to induce NGN2 expression via the Tet operon. A complete media change was performed on day 1. On day 2, NPCs were detached from the plate using Accutase (Gibco) and re-seeded in plates pre-coated with poly-L-ornithine (0.01%, Sigma) and Matrigel (20 ug/ml, Corning) at 75,000–125,000 cells/cm$^2$ in B27-plus/Neurobasal plus media containing 10 ng/ml BDNF (R&D), 10 ng/ml GDNF (R&D), 200 uM dcAMP (Sigma), 200 uM ascorbic acid (Sigma), 1 ug/ml laminin (Sigma) (B27-Plus Complete media), plus 2 ug/ml DOX, 500 nM RO4929097 (Roche), and 10 uM Y-27632 (Calbiochem). On day 3, media was replaced with B27-Plus Complete containing DOX and RO4929097. On day 6, media was replaced with B27-Plus Complete without DOX and RO4929097. Thereafter, 50% media changes with B27-Plus Complete were performed every 4–5 days. iNs were assessed at day 23 unless otherwise noted.

### Immunocytochemistry

Immunofluorescence was performed on day 23 iNs in Cell Carrier-96 Ultra microplates (Perkin Elmer). Cells were fixed in 4% formaldehyde at room temperature for 15 min. Fixed cells

**Table 1. Antibodies used in immunofluorescence.**

| Antigen | Company/Cat # | Dilution | Secondary | Secondary Company/Cat # |
|---|---|---|---|---|
| MAP2 | Abcam ab5392 | 1:5000 | Gt anti-Ch AF488 | Abcam ab150169 |
| | | | Gt anti-Ch AF568 | Abcam ab175477 |
| Tau CP27 | Davies Lab [17] | 1:500 | Gt anti-Ms IgG2b AF647 | Thermo A-21242 |
| CUX1 | Abcam ab54583 | 1:100 | Gt anti-Ms IgG1 AF488 | Thermo A-21121 |
| SOX2 | Abcam ab97959 | 1:500 | Gt anti-Rb AF568 | Abcam ab175471 |
| TUJ1 | Biolegend 801201 | 1:1000 | Gt anti-Ms IgG2a AF647 | Thermo A21241 |

were blocked and permeabilized in 3% BSA, 0.3% Triton X-100 in PBS at room temperature for 45 min and incubated with primary antibodies (Table 1) in block/perm buffer overnight at 4˚C. The next day, cells were incubated with secondary antibodies (Table 1) diluted 1:500 in block/perm buffer at room temperature for 1–1.5 hr. Nuclei were counterstained with 1 uM DAPI in PBS. Images were collected on the Operetta CLS with Harmony software (Perkin Elmer).

## RNA isolation

RNA was isolated from hiPSC-derived iNs at days 0, 2, 6, 13, and 23 using the AllPrep DNA/RNA mini kit (Qiagen). Samples were extracted manually according to the manufacturer's instructions or on a QIAcube using the AllPrep DNA/RNA mini kit protocol (Qiagen).

## Reverse transcription and quantitative PCR

cDNA was generated with SuperScript VILO Master Mix (Thermo) using 100 ng—1 ug input RNA according to the manufacturer's instructions.

High-throughput qPCR was performed using the Biomark HD system (Fluidigm) according to the manufacturer's instructions. Briefly, a custom 96-gene Delta Gene Assay panel (Fluidigm) was designed (S1 Table) and pooled to generate a stock containing 500 nM final concentration of each assay. The pooled assays were used in a 10 ul preamplification reaction with 2.5 ul cDNA and 2X TaqMan Preamp Master Mix (Applied Biosystems) under the following conditions: 95˚C for 10 min; 10 cycles of 95˚C for 15 sec, 60˚C for 4 min. Preamp reactions were cleaned up with 16 U of Exonuclease I (NEB) in 1X Exonuclease I reaction buffer (NEB) at 37˚C for 30 min, then heat inactivated at 80˚C for 15 min. Samples were diluted 5-fold, combined with Pre-Mix (1X SsoFast EvaGreen Supermix (Biorad), 1X DNA binding dye (Fluidigm)), and 5 ul was pipetted into the sample inlets on a primed 96.96 IFC (Fluidigm). The assay mix was prepared with 5 uM final concentration of each Delta Gene assay and 1X Assay Loading Reagent (Fluidigm), and 5 ul was pipetted into the assay inlets of the IFC. The IFC was loaded using the Juno system (Fluidigm) and data was collected on the Biomark HD system (Fluidigm) using the GE Fast 96x96 PCR+Melt v2.pcl program. Data was analyzed using the Fluidigm Real-Time PCR analysis software (Fluidigm SC) and custom R scripts (ggplot2, d3heatmap).

Standard qPCR assays were performed using PowerUp SYBR Green (Applied Biosystems) according to the manufacturer's instructions and cycled on a CFX384 instrument (Biorad) under the following conditions: 50˚C for 2 min, 95˚C for 2 min; cycled 40x at 95˚C for 15 sec, 60˚C for 1 min. Cq values were normalized against the geometric mean of three housekeeping genes (*ACTB, B2M, GAPDH*). Refer to Table 2 for primers used.

**Table 2. Primers used in SYBR qPCR.**

| mRNA | F primer | R primer |
|------|----------|----------|
| ACTB | CACCAACTGGGACGACAT | ACAGCCTGGATAGCAACG |
| B2M | CTCTCTCTTTCTGGCCTGGAG | TCTGCTGGATGACGTGAGTA |
| GAPDH | GGATTTGGTCGTATTGGG | GGAAGATGGTGATGGGATT |
| FOSB (total) | TCTGTCTTCGGTGGACTCCT | AGGTCCTGGCTGGTTGTGAT |
| ΔFOSB | GCAGAGCTGGAGTCGGAGAT | GCCGAGGACTTGAACTTCACTCC |
| ERCC1 (total) | CAACCTGCACCCAGACTACA | AGTCGGCCAGGATACACATC |
| ERCC1 long | CAAAGGACGCGACCATAACT | CGTCGAGGGGTATCACAAAT |
| CAT | ACCGAGAGAGAATTCCTGAGA | GCCTTGGAGTATTTGGTAATGTC |
| PCDHB5 | CAAACTCTAAAAGAAGGACGCAT | AATAACCGCGAATCCGATCT |

## RNA sequencing

For each timepoint, RNA from 4 technical replicates (4 independent wells of cells cultured simultaneously) were used per line. RNA library preparation from total RNA was conducted following the manufacturer's protocol for the Kapa mRNA HyperPrep Kit (Roche). Briefly, 250 ng of total RNA was enriched for mRNA using magnetic oligo-dT beads. The remaining RNA was then fragmented by magnesium under elevated temperature. After fragmentation, RNA was depleted of rRNA and globin mRNA using the QIAseq FastSelect RNA Removal Kit (Qiagen). The depleted RNA then underwent first strand synthesis using reverse transcriptase and random primers. Combined second strand synthesis and A-tailing incorporated dUTP into the second cDNA strand for stranded RNA sequencing and added dAMP to the 3' ends for adapter ligation. The cDNA fragments were then ligated to sequencing adaptors (IDT xGEN Dual Index UMI adapters) and was enriched using 16 cycles of PCR. Final libraries were assessed using the Agilent Tapestation and Qubit (ThermoFisher) assay methods then sequenced on an Illumina HiSeq 4000 sequencer using 2 x 75bp read length.

Raw fastq reads were subjected to quality control using FastQC (v0.11.7) and quality profiling, filtering for low quality and short reads, adapter trimming and mismatch correction using fastp (v0.20.1) [18]. RNA reads were aligned to the hg38 build of the reference human genome, with transcript annotations from GENCODE, using the STAR aligner. Gene expression was quantified using RSEM (RNA-Sequencing by Expectation Maximization) [19]. The expected count matrix from RSEM was filtered for low expression using a counts-per-million threshold of 1 in at least 40 samples (number of samples in the smallest group of comparison) and further filtered for genes annotated as pseudogenes or pseudo-autosomal regions. A linear model was fit on the TMM-voom normalized data using the limma (v3.44.3) package in R [20]. Differential gene expression analysis was conducted using an empirical Bayes statistic in limma for the genotype x day variable of interest [21]. Raw p-values were adjusted for multiple comparison tests using the Benjamini-Hochberg correction and an adjusted p-value threshold of 0.05 and absolute $\log_2 FC > 1$, was used to identify differentially expressed genes.

## Pathway and gene set enrichment analysis

Results from RNA-seq studies were analyzed using Ingenuity Pathway Analysis (IPA, Qiagen Inc., https://www.qiagenbioinformatics.com/products/ingenuitypathway-analysis) using the core analysis and comparison analysis functions and cutoffs of FDR < 0.05 and $\log_2 FC < -1$ & > 1. Venn diagrams were generated using http://bioinformatics.psb.ugent.be/webtools/Venn/.

Overrepresented pathways were further identified (q-value < 0.01) using the Reactome database and visualized using the cnetplot via clusterProfiler (v3.16.1) in R [22].

## Amyloid beta and cell viability assays

To align media conditions prior to collection, a complete media change (B27-Plus Complete) was performed on day 22 iNs grown in 96-well plates. Control wells were treated with DMSO vehicle or 10 uM DAPT gamma-secretase inhibitor as a negative control for Aβ production. On day 23, conditioned media was collected from cells and ELISA for amyloid beta was performed using the V-Plex Aβ Peptide Panel 1 (6E10) kit (Meso Scale Discovery) according to the manufacturer's instructions. Cells remaining in each well were analyzed by CellTiter-Glo™ (Promega) according to the manufacturer's instructions to determine relative cell viability. MSD Aβ values were normalized to standard curves for each protein isoform and relative fluorescence units obtained from CellTiter-Glo™.

## Western blotting

Cells were lysed in Mammalian Protein Extraction Reagent (Thermo) supplemented with protease inhibitor cocktail (Roche). Lysates were separated on NuPAGE 4–12% Bis-Tris protein gels (Invitrogen) in NuPAGE MOPS SDS Running buffer (Invitrogen). Protein was transferred to PVDF membranes using the Trans-Blot Turbo System (Biorad). Membranes were blocked in Intercept Blocking Buffer (Licor) at room temperature for 1 hr and incubated overnight in primary antibodies diluted in 50% Intercept Blocking Buffer, 50% PBS, 0.1% Tween-20. See Table 3 for primary antibodies used. Donkey anti-rabbit IgG IRDye 800CW secondary antibody (Licor) was diluted 1:10,000 in 50% Intercept Blocking Buffer, 50% PBS, 0.1% Tween-20 and incubated with membranes at room temperature for 1 hr. Membranes were imaged using the Odyssey CLx (Licor) and quantified using ImageJ software (NIH).

## Calcium imaging

AAV6-syn-GCaMP6f purchased from Vigene Biosciences was diluted with NMM and applied to the differentiated hiPSC-neurons on day 7 to achieve 25k MOI (multiplicity of infection). After day 20, calcium imaging was performed by a high-content imaging machine (PerkinElmer Opera Phenix). With the longitudinal imaging, we determined that day 45 showed good level of synchronous firing. For each imaging, 2000 frames of images were continuously acquired at 3 Hz. During imaging, temperature and $CO_2$ were maintained at 37°C and 5%, respectively. The raw images were processed by a constrained nonnegative matrix factorization (CNMF) algorithm [23] to determine whether individual neurons were fired at a given time point and to visualize the firing events in a raster plot. Then, the raster plot was transformed into cofiring ratio plot (number of fired neurons at a given time / number of total neurons). Finally, the coactivity plots were analyzed in terms of peak amplitude and frequency to evaluate the level of functional circuit formation.

**Table 3. Antibodies used in western blotting.**

| Antigen | Company/Cat # | Dilution |
|---------|---------------|----------|
| GAPDH | Cell Signaling Technology, #5174 | 1:5000 |
| FOSB | Cell Signaling Technology, #2251 | 1:1000 |
| Catalase | Cell Signaling Technology, #12980 | 1:1000 |

## Conditional GWAS analysis

Conditional analysis was performed using the COJO approach [24] (with the—*cojo-cond* command) implemented in the GCTA software (v1.93.2) [25]. LD structure within 1 Mb in each direction of the conditioning variant rs429358 was estimated using imputed genotype data from the entire FinnGen cohort. Full summary statistics from the wide definition AD GWAS obtained from the FinnGen project database were included in the conditional analysis.

## Statistics

Unless otherwise noted, results are expressed as means +/- standard deviation. Statistical methods used to analyze RNA-seq data are described in the "RNA sequencing" section above. Other statistical tests were performed using Prism software (Graphpad) and are described in their respective figure legends.

# Results

## Identification of a rare variant associated with AD susceptibility

The FinnGen study database contains genetic information and disease phenotype data from individuals of Finnish descent, a population displaying a founder effect and therefore possessing greater potential to pinpoint disease-associated variants compared to more heterogeneous populations. Finland has one of the highest rates of Alzheimer's disease mortality worldwide [13]. To identify novel AD-associated variants that are specifically enriched in Finns and that may contribute to this high AD prevalence, we queried the PheWAS results on the wide definition of Alzheimer's disease phenotypes (FinnGen, data freezes (DF) 3–6; https://r6.finngen.fi/ [12]) (Fig 1). This analysis revealed a novel variant, rs148726219, located on chromosome 19q13.32 (>500 kb from *APOE*) which showed highly significant association with AD in DF3 (p = 8.9e-34, OR = 2.5) and had further increased significance in the later release DF6 (p = 2.7e-54). Importantly, rs148726219 significantly associated with alternative definitions of Alzheimer's disease, other neurodegenerative disorders, and dementia in the Finnish population (S1 Fig). Interestingly, the low-frequency (3.7%) risk allele T of rs148726219 appears to have arisen on the common haplotype tagged by the more common and well-established (18%) risk allele C of rs429358 in *APOE*, with which it is in linkage disequilibrium (D' = 0.77) but weakly correlated ($r^2$ = 0.122) with. A conditional association test on rs429358 revealed that rs148726219 lost its genome-wide significance (ORcond = 1.198445 (95% CI 1.031881–1.391896); Pcond = 0.0177376), suggesting that there are two risk alleles on the common, ancestral rs429358 risk haplotype. This is not unexpected given the emerging evidence in the literature, supported by functional studies, of multiple tightly linked causal SNPs within the same locus contributing to disease risk [26]. With clear functional consequences outlined in this report, the novel SNP that we have identified here provides another example for evidence of multiple causal SNPs at a single GWAS locus.

Compared to other nearby SNPs from DF3, rs148726219 is highly enriched (136-fold) and relatively common (minor allele frequency = 3.7%) in the Finnish population (Fig 1), and rare in other populations (1000Genomes 0.2%, TOPMED 0.04%, gnomAD 0.6%), suggesting that rs148726219 may be an AD-associated risk variant enriched by the genetic and population bottlenecks of the Finish population and occurring on the same haplotype as older AD-associated variants such as rs429358. Importantly, carriers of both the rs148726219 and rs429358 variant risk alleles (heterozygous or homozygous states) were at higher risk of AD phenotype diagnosis (OR = 2.44) compared to individuals with either rs148726219 or rs429358 alleles alone (OR = 1.6 and OR = 2.32 respectively) underscoring the combined risk profile of both SNPs.

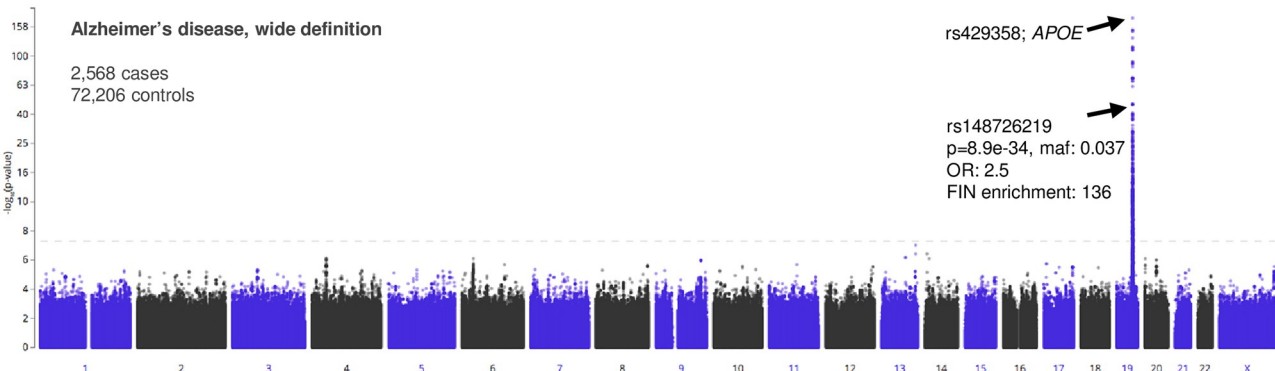

**Fig 1. FinnGen PheWAS identifies an AD-associated intronic susceptibility variant enriched in the Finnish population.** Manhattan plot from FinnGen phenome-wide association study (PheWAS, data freeze 3) of Alzheimer's disease, using the wide definition (ICD-10 codes G30 & F00) [12]. The SNP of interest (rs148726219) is shown, which is located ~600 kb away from the lead SNP (rs429358) at the *APOE* locus. maf = minor allele frequency; OR = odds ratio; FIN enrichment = enrichment score in Finnish population (where a score of 1 is neutral).

Based on its strong statistical significance in FinnGen DF3 and DF6 and phenotypic associations presented above, and in order to better characterize other risk variants outside of *APOE* (rs429358) that may underlie the association of this region with AD, rs148726219 was chosen for further mechanistic assessment to determine the functional consequences of this novel variant.

### Generation of rs148726219-containing hiPSC lines

In order to avoid *APOE*-associated effects, we opted to edit the genotype of rs148726219 in its gene locus instead of ablating the LD block it resides in. rs148726219 is located within intron 2 of the *FOSB* gene (ENST00000353609.8), a member of the AP-1 complex which has an exceptionally stable splice isoform *(ΔFOSB,* ENST00000615753.4, ENST00000592436.5) that has been previously implicated in addiction, epilepsy, and cognition [27–29]. Also spanning the locus is an unstudied isoform of *ERCC1*, a component of the nucleotide excision repair pathway, that utilizes a cryptic upstream promoter (*"ERCC1-long"*, ENST00000423698.6, Fig 2A). To begin to functionalize rs148726219 in AD, we utilized CRISPR/Cas9-mediated homology

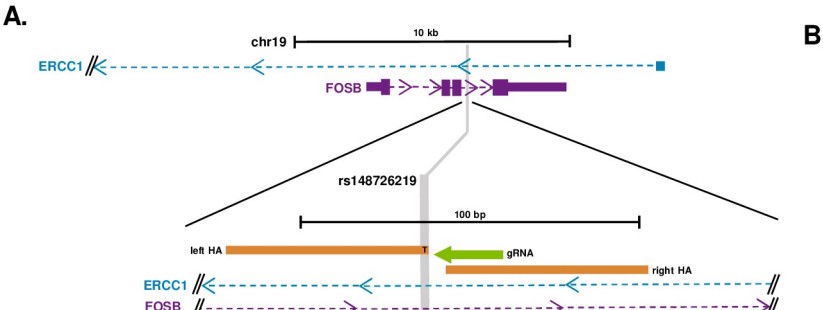

**B.**

| hiPSC edited clones | rs148726219 status | Variant allele frequency |
|---|---|---|
| WT – 2A1 | C/C | 0.00 |
| Het – 2D2 | C/T | 0.54 |
| Het – 2G6 | C/T | 0.56 |
| Hom – 2B11 | T/T | 0.71 |
| Hom – 2H6 | T/T | 0.79 |

**Fig 2. SNP-editing of rs148726219 using CRISPR/Cas9 produces clonal lines with heterozygous and homozygous alleles. A.** Schematic of the targeting strategy for rs148726219 using CRISPR/Cas9 and HDR using homology arms flanking the PAM site within the *FOSB* locus. The left homology arm (left HA) provided the template for the C>T edit. **B.** Isogenic clones were generated in the BIONI010-C-13 hiPSC line to contain wild type (WT), heterozygous (Het), or homozygous (Hom) alleles of rs148726219. Clone genotypes were initially confirmed by Sanger sequencing; variant allele frequency was determined by whole genome sequencing.

directed repair in the BIONi010-C-13 hiPSC line [14] (derived from an individual of African-American descent) combined with single cell cloning to engineer hiPSC clones that contained heterozygous (HET) or homozygous (HOM) alleles of the SNP (Fig 2A and 2B). Two HET (HET-2D2, HET-2G6) and 2 HOM (HOM-2B11, HOM-2H6) clones were generated, and genotypes were confirmed by NGS sequencing, along with a wild type, untargeted clone harboring the non-risk allele, (WT-2A1; herein referred to as 'WT') from the same experiment (Fig 2B). For all lines, karyotypes by G-banding and STR profiling were normal as compared to the parental hiPSC line (S2 Fig).

## rs148726219-edited hiPSC lines do not exhibit apparent defects in neuronal differentiation or neuron-like morphology

The BIONi010-C-13 line contains a DOX-inducible *NGN2* transgene inserted into the AAVS1 safe-harbor locus and can be rapidly converted into cells of neuronal lineage, with functioning neurons generated after just 2 weeks [14, 15]. In at least 3 separate differentiation campaigns, we performed forced *NGN2* overexpression in the rs148726219-edited BIONi10-C-13 lines in parallel with the WT and parental lines (both non-targeted genotype) and differentiated the induced neurons (iNs) for 23 days post-DOX-induction to allow for robust synapse development (Fig 3A and 3B). At day 23, iNs expressed pan-neuronal markers MAP2 and Tau nearly ubiquitously, while apparent morphology did not differ between lines (Fig 3C). TUJ1, another pan-neuronal marker, and CUX1, indicating cortical layer II-IV, were also highly expressed by all the lines; undifferentiated hiPSCs or neural progenitor cells (NPCs) marked by SOX2 were not apparent (S3A Fig). Moreover, the number of cells at day 23 was not substantively different between lines (S3B Fig) demonstrating no consequence of the gene editing process on viability. These data suggest that hiPSC-iN differentiation occurred equally and robustly across all CRISPR-edited lines.

To further ensure an absence of more subtle defects in hiPSC-iN differentiation, we also assessed mRNA expression by high-throughput qPCR of rs148726219-edited lines at day 0, 2, 6, 13, and 23 post-DOX induction using a panel of 96 cell type-specific markers (S1 Table). Hierarchical clustering and principal component analysis (PCA) showed that samples clustered by timepoint, not by clone or by rs148726219 genotype (Fig 3D, S4A Fig). This data indicates that major gene expression changes identified in this 96 gene panel are driven by hiPSC-iN differentiation and not by inherent differences introduced by CRISPR or through the generation of clones from single cells. By analyzing the high throughput qPCR data relative to the geometric mean of 3 housekeeping genes (*GAPDH*, *EIF4A2*, *RPL13)*, we found that *NGN2* expression was increased with DOX induction, as expected, and reduced after treatment cessation in all lines. Expression at day 0 (prior to DOX induction) was essentially zero, indicating no transgene leakiness (S4B Fig), and no differences in *NGN2* expression were detected based on genotype or clone (S4C Fig). As expected, markers of mature neurons increased throughout iN differentiation (*DCX*, *DLG4*, *MAP2*, *MAPT*, *RBFOX3*, *SLC17A6*, *SLC17A7*, *SYN1*, *SYP*, *TUBB3*), but none of these were significantly different across lines (S5A Fig). Markers of pluripotency (*LIN28A*, *MKI67*, *NANOG*, *POU5F1*; S5B Fig) and NPCs (*DACH1*, *OTX2*, *PAX6*, *PROM1*, *SOX2*; S5C Fig*)* were not expressed in mature iNs, and non-neuronal markers (*CDX2*, *GATA4*, *GFAP*, *OLIG1*, *S100B*, *SLC1A3*, *SOX17*, *TBXT*; S5D Fig*)* were not detected past the progenitor cell stage in any line. Furthermore, no differences in expression of AD-associated markers, including *APOE*, the 4R isoform of *MAPT*, *PSEN1*, *APP* (S5E Fig), or secreted amyloid beta (Aβ) (S6 Fig), were detected across any of the edited lines. Collectively, these observations underscore that hiPSC-iN conversion occurred equally across all rs148726219-edited, WT, and parental BIONi010-C-13 lines. Based on the equal potential of

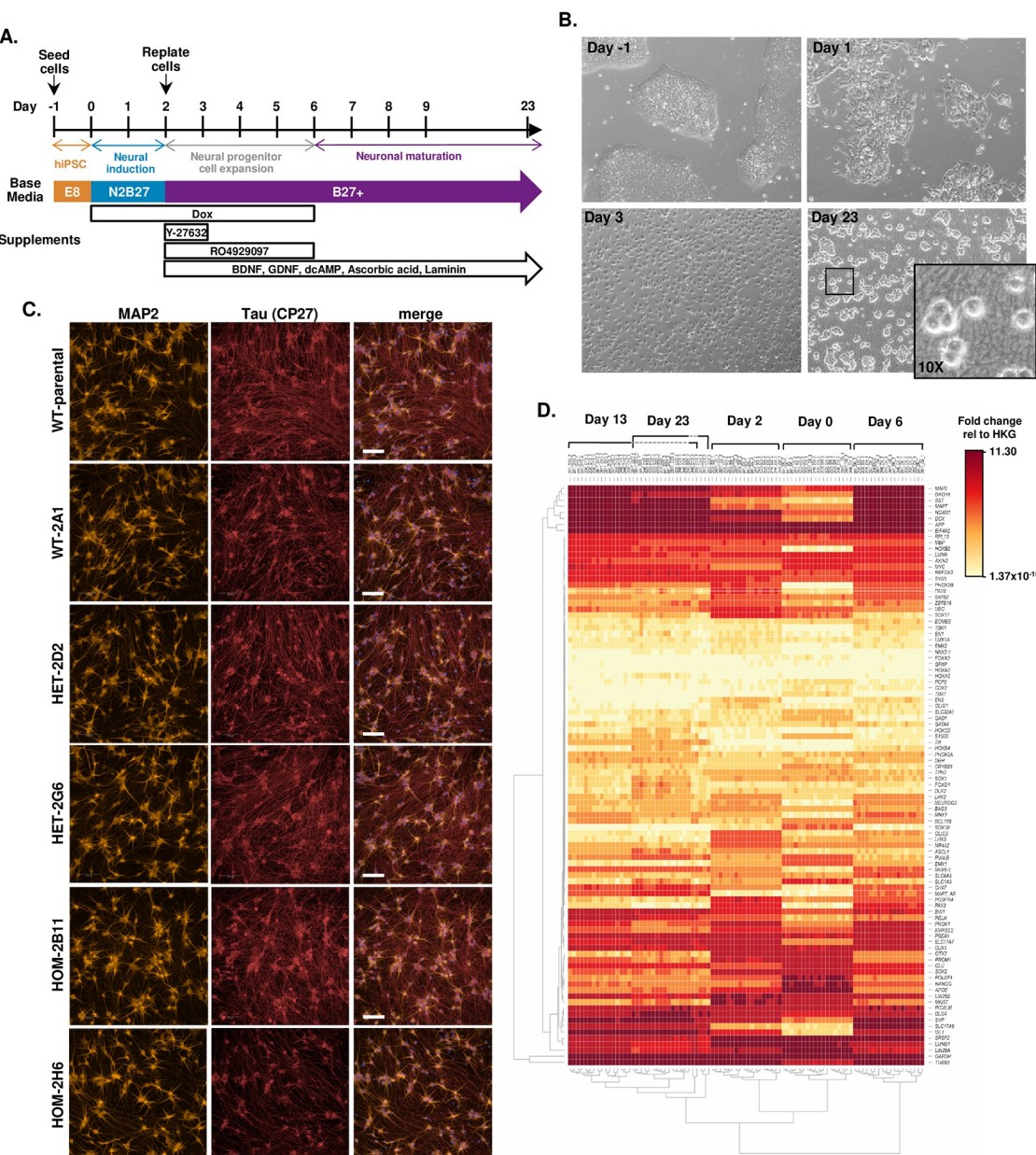

**Fig 3. hiPSCs and iNeurons from SNP-edited lines show no differences in differentiation potential. A.** hiPSC-iNeuron differentiation protocol utilizing the doxycycline-inducible neurogenin 2 (iNGN2) transgene. **B.** Representative images of BIONi010-C-13 cells at different time points during iN differentiation. **C.** Immunocytochemistry of day 23 iN indicates no differences between clones in cell morphology or levels of neuronal marker proteins. Representative images of MAP2 (orange) and Tau (red) are shown for the BIONi010-C-13 parental line, and WT-2A1, HET-2D2, HET-2G6, HOM-2B11, and HOM-2H6 clones. The merged images include DAPI nuclear counterstain (blue). Scale bars = 100 um. **D.** Heatmap of high-throughput qPCR data from BIONi010-C-13 parental, WT-2A1, HET-2D2, HET-2G6, HOM-2B11, and HOM-2H6 lines at days 0, 2, 6, 13, and 23 of 1 representative iN differentiation. Values were normalized relative to the geometric mean of 3 housekeeping genes (HKG; *GAPDH*, *EIF4A2*, *RPL13*) and expressed as fold change values (ΔCt). Three technical replicates of each line are shown per timepoint. Samples cluster by timepoint, not by line. Some overlaps exists between samples on day 13 and 23.

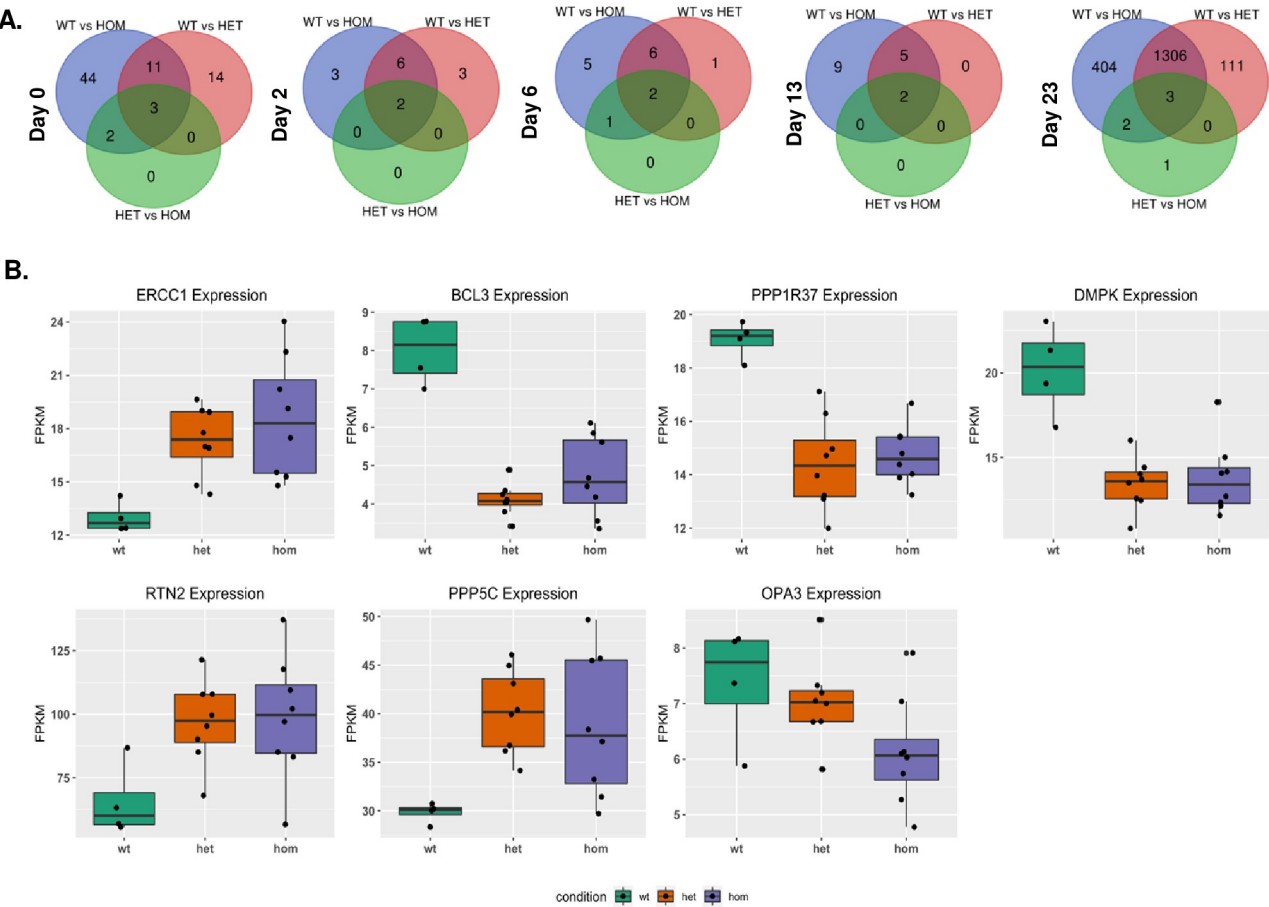

**Fig 4. RNA-seq reveals the effects of rs148726219 on proximal gene expression. A.** Venn diagrams indicating the number of differentially expressed genes (DEGs) identified by RNA-seq between heterozygous clones (HET-2D2, HET-2G6), homozygous clones (HOM-2B11, HOM-2H6), and the wild type clone (WT-2A1) at days 0, 2, 6, 13, and 23 of hiPSC-iNeuron differentiation. Significant genes were determined by FDR <0.05 and $\log_2$FC values <-1.0 and >1.0. **B.** Genes that are within 1Mb of rs148726219 and significantly differentially expressed in day 23 iN cells between WT (green bars), HET (orange bars) and HOM lines (purple bars) based on FDR <0.1 and $\log_2$FC between -0.5 and 0.5. Expression values are expressed as fragments per kilobase per million mapped reads (FPKM) at day 23 of hiPSC-iNeuron differentiation. Four technical replicates for each clone are shown with each represented by a black dot.

all genotypes to form hiPSC-iN cells, we were able to confidently investigate the biological impact of this variant.

## RNA-seq identifies transcriptome-wide effects of rs148726219 in mature iN function and survival

To begin functional characterization, we next performed RNA sequencing (RNA-seq) in the rs148726219-edited clones, the isogenic WT, and the parental line. All 6 hiPSC lines, characterized above, were differentiated to iNs in parallel and RNA was extracted in quadruplicate on days 0, 2, 6, 13, and 23 post-DOX-induction. Libraries were generated and RNA-seq performed, which revealed sample clustering by timepoint alone and not by genotype, similar to the PCA results observed by high throughput qPCR in S4A Fig (S7A and S7B Fig). No obvious differences in clustering were detected between the BIONi010-C-13 parental line and the WT clone (S7C Fig) so the WT clone was used for further differential gene expression analysis.

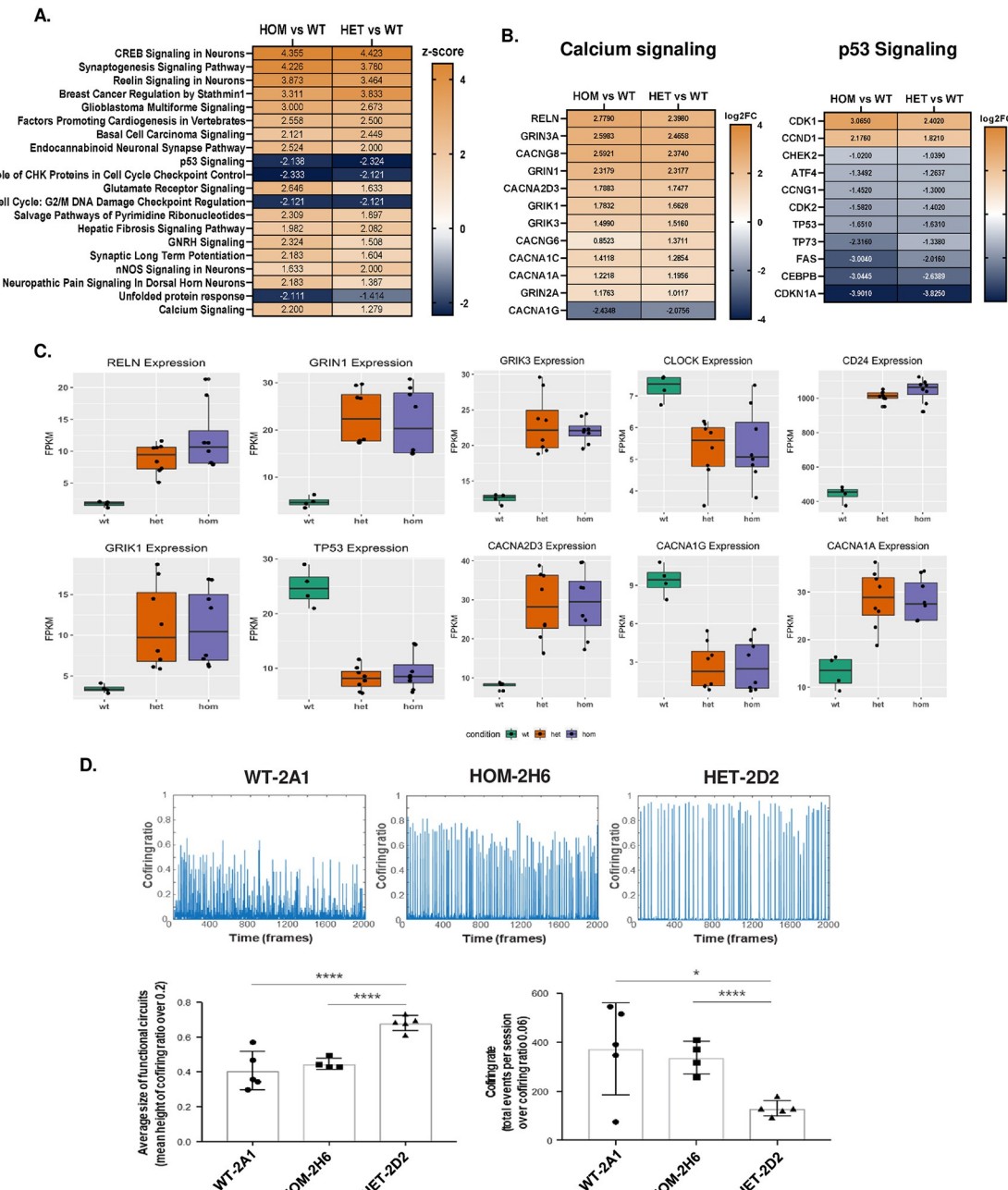

**Fig 5. AD-relevant pathways and neuron circuit firing are differentially regulated in rs148726219-edited mature iNeurons. A.** IPA gene ontology comparison analysis showing top 20 dysregulated pathways between WT-2A1 and homozygous clones (left column) and WT-2A1 and heterozygous clones (right column) on day 23. Z-scores denote level of pathway upregulation (orange) or downregulation (navy) based on $\log_2 FC$ values of DEGs contained within them. **B.** Differentially expressed genes involved in synaptogenesis, calcium signaling, and glutamate signaling (left), and cell cycle and unfolded protein response (right) based on FDR $<0.05$, $\log_2 FC <-1.0$ and $>1.0$. Color denotes $\log_2 FC$ values; orange = upregulated, navy = downregulated. **C.** Differentially expressed genes in day 23 iN cells identified by RNA-seq involved in synaptogenesis, calcium signaling, glutamate signaling, cell cycle, and unfolded protein response based on FDR $<0.05$ and $\log_2 FC$ between -1.0 and 1.0. Expression values are expressed as fragments per kilobase per million mapped reads (FPKM) for WT (green bars), HET (orange bars) and HOM lines (purple bars). Four technical replicates for each clone are shown with each represented by a black dot. **D.** Calcium imaging from 1 representative experiment shows rs148726219-edited heterozygous hiPSC lines to have significantly enhanced functional circuit maturation compared to wildtype and homozygous lines. Top: Level of functional circuit activity and cofiring ratio was calculated (number of neurons fired at a time point over total excitable neurons). Greater peak amplitude of cofiring ratio of HET-2D2 line implicates formation of bigger circuits. Bottom: Chart of mean peak amplitude in 5 individual replicates (each represented by a dot) of the HET-2D2 line shows statistically significant increase of functional circuit size compared to WT-2A1 and HOM-2H6 lines. Also,

significantly less frequent circuit firing was observed with larger circuit formation in HET-2D2 line. * $p < 0.5$, **** $p < 0.0001$ by one-way ANOVA with multiple comparisons.

Using significance thresholds of false discovery rates (FDR) $<0.05$ and gene expression $\log_2$ fold changes ($\log_2 FC$) $<-1$ & $>1$, we identified several DEGs between HET, HOM, and WT clones across all timepoints (Fig 4A, S8 and S9 Figs). The greatest differences were observed between HET or HOM clones compared to WT, though 2–6 genes (depending on timepoint) were also significant between HET and HOM lines. Sixty or fewer DEGs were identified across all comparisons at day 0, 2, 6, and 13, while on day 23, HET and HOM iNs each had approximately 1500 DEGs compared to WT. The majority of DEGs in HET clones were also differentially regulated in HOM clones compared to WT.

Due to the location of rs148726219 within introns of *FOSB* and *ERCC1-long*, we initially queried whether their transcripts showed altered expression in CRISPR-edited iNs. *FOSB*

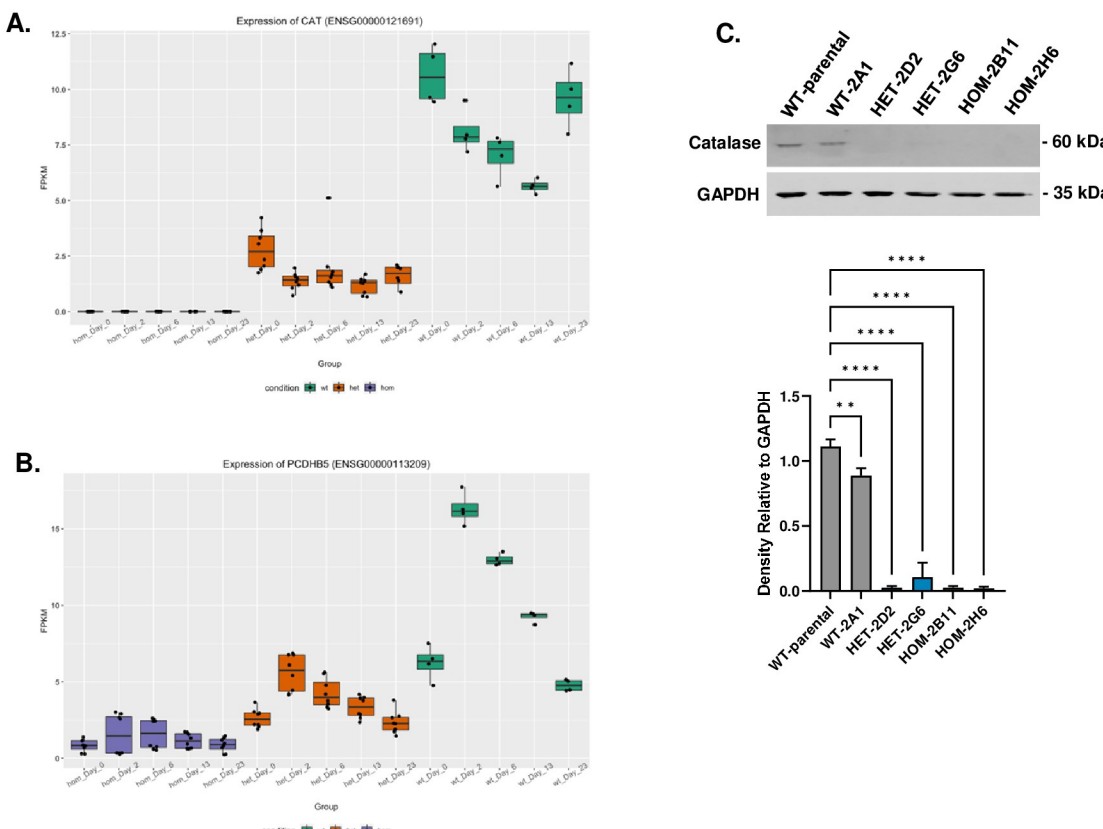

**Fig 6. *CAT* and *PCDHB5* are differentially expressed in rs148726219-edited cells throughout hiPSC-iN differentiation. A.** *CAT* expression measured by RNA-seq shown as fragments per kilobase per million mapped reads (FPKM) for WT-2A1 (green), rs148726219-heterozygous (orange), and homozygous (purple) clones at day 0, 2, 6, 13, and 23 of hiPSC-iNeuron differentiation. Four technical replicates for each clone are shown with each represented by a black dot. **B.** *PCDHB5* expression measured by RNA-seq, as described in (A). **C.** Representative western blot of day 23 iNeuron lysates in BIONi010-C-13 parental, WT-2A1, HET-2D2, HET-2G6, HOM-2B11, and HOM-2H6 lines probed with an antibody against catalase (top). Protein was quantified relative to GAPDH loading control and is expressed as fold change relative to the average of WT-parental and WT-2A1 lines (bottom; additional WBs used for quantification are shown in S19 Fig). n = 3. ** $p < 0.01$, **** $p < 0.0001$ by one-way ANOVA with multiple comparisons.

mRNA was not detected in our RNA-seq data at iPSC or iN stages in any of the clones due to low counts per million (CPM) values. This low expression agrees with gene expression data from brain-relevant regions in the GTEx database (TPM ~<4) [30]. Furthermore, we did not find differences in *FOSB* or the *ΔFOSB* splice variant by qPCR-based analysis of multiple independent experiments, although both transcripts were so lowly expressed (Ct > 35) in most samples that they could not be reliably quantified (S10A and S10B Fig). To determine whether induction of the *FOSB* gene could reveal differences in expression between rs148726219-edited clones, iNs were stimulated for 4 hours with phorbol 12-myristate 13-acetate (PMA), a known inducer of *FOSB* and other members of the AP-1 complex [31] (S11 Fig). Though protein levels were clearly induced with PMA, no apparent differences in total FOSB expression were detected between rs148726219-edited clones and the WT clone; baseline expression of ΔFOSB in the absence of PMA showed subtle, non-significant increases in the HOM clones which warrants further investigation. On the other hand, *ERCC1* was highly detectable by RNA-seq (FPKM ranging ~10–55, S12A Fig). After an initial increase in expression on day 2, iN maturation corresponded to gradual reductions in *ERCC1* levels. On day 23, *ERCC1* levels were significantly increased by RNA-seq ($\log_2$FC >0.5 and FDR <0.1) in rs148726219-edited HET and HOM clones compared to WT (Fig 4B); however, qPCR-based analysis of *ERCC1* and the *ERCC1-long* isoform from independent experiments did not show measurable differences between genotypes (S10C and S10D Fig). Therefore, in iNs, rs148726219 does not strongly impact expression of the genes it is harbored within. However, we did detect subtle effects of the SNP on ΔFOSB and *ERCC1*, and it remains possibile that rs148726219 could influence their expression more strongly in other cell types.

To probe the effect of rs148726219 on expression of other neighboring genes in *cis*, we next assessed expression of proximal upstream and downstream genes within 1 Mb spanning rs148726219 (hg38 chr19: 44,971,732–45,971,732) that were detectable by RNA-seq and differentially expressed between WT and HET/HOM clones. Of the expressed genes that met these criteria, *BCL3*, *PPP1R37*, *RTN2*, *OPA3*, *EML2*, and *GEMIN7* significantly differed in either direction with $\log_2$FC of at least 0.5 and FDR <0.1 in day 23 iN cells (Fig 4B, S12B–S12D Fig; S2 Table). Other proximal genes influenced by rs148726219, albeit with lower $\log_2$FC ranges, were *DMPK*, *PPP5C*, *PVR*, *PNMA8A*, *BLOC1S3*, *ZNF180*, *FBXO46* and *ERCC2* (Fig 4B, S12E Fig; S2 Table). We did not observe these genotype-driven differences in earlier timepoints of iN differentiation. Due to the proximity of rs148726219 to *APOE* (~600 kb), we also probed its expression from our RNA-seq data (S12F Fig). Expectedly, we observed almost no *APOE* transcript expression in mature iNs [32], and relatively small differences between genotypes. These observations suggest a weak, but plausible regulatory mechanism for rs148726219 on select upstream and downstream neighboring genes evidenced in mature iN cells that is worthy of further interrogation.

In addition to these small, but significant, *cis* effects of rs148726219, hundreds of DEGs were detected between HET/HOM and WT iNs at day 23, when iN cells had reached an early maturation timepoint. Therefore, we next performed pathway overrepresentation analysis to determine biologically relevant pathways that were altered at this timepoint. The Ingenuity Pathway Analysis (IPA) tool revealed several neuron-relevant signaling pathways as upregulated in both HET and HOM clones compared to WT, including Calcium Signaling, Wnt/β-catenin, and Axonal Guidance System (S13A and S13B Fig). Comparison analysis revealed CREB Signaling, Synaptogenesis Signaling Pathway, and Reelin Signaling in Neurons to be the top upregulated pathways in both HET and HOM clones, along with Glutamate Receptor Signaling and Calcium Signaling (Fig 5A). These pathway enrichments were largely driven by upregulation of several neuron-specific factors, such as *RELN* (Reelin), calcium channel subunits (*CACNA1A*, *CACNA1C*, *CACNG8*, *CACNA2D3*, *CACNG6)*, and glutamate receptor

subunits (*GRIN3A*, *GRIN1*, *GRIK1*, *GRIK3*, *GRIN2A*) (Fig 5B and 5C, S14 and S15 Figs). Several genes that could have implications in neurogenesis were also upregulated, including Fibroblast growth factor receptor *(FGFR2)* and Transforming growth factor beta *(TGFB2)* (S15 Fig) [33, 34]. A second computational method of pathway enrichment, Cnetplot, also identified Neuronal System and Extracellular Matrix-related pathways, including members of the TGF-β/BMP and integrin families, as upregulated in HET and HOM clones compared to WT (S16 and S17 Figs). The extracellular matrix (ECM) in particular is a known regulator of neuronal morphology and function, and its dysregulation is manifested in AD via blockage of synaptic transmission and elimination of amyloid plaques including via Integrin α5β1 complexes [35, 36]. We observed *ITGA5* (Integrin α5) upregulation in HET and HOM clones suggestive of a dysregulation of its interaction with ECM of neurons. These genes were only upregulated at day 23—not throughout hiPSC-iN differentiation—and we did not observe evidence of contaminating immature or non-neuronal cell types in our samples (S3 and S5 Figs); therefore, these results suggest that the presence of rs148726219 mediates important biological effects only on mature neuron function and signaling.

The predicted functional effect of rs148726219 in synaptogenesis and neuronal circuit maturation was confirmed by optical electrophysiology. To evaluate functional circuit maturation, calcium imaging of three AAV6-syn-GCaMP6f induced lines (WT-2A1, HET-2D2, and HOM-2H6) was performed on day 45 after iN synapses reach further levels of maturity (Supplemental Movie). This analysis showed that the HET-2D2 line had more neurons participating in synchronous firing compared to the WT-2A1 and HOM-2H6 lines as shown in cofiring ratio plots (Fig 5D, top panels) and raster plots (S18 Fig). More bars lined up in raster plots and higher peak amplitude in cofiring ratio plots indicates bigger and more mature functional circuits. Higher frequency circuit cofiring is indicative of fragmented circuits, and this was observed in the WT line, and to a lesser extent, the HOM line, which, although not significant, trended toward an increase in synchronous firing compared to WT (Fig 5D, bottom panels). These results showing the rs148726219 HET clone to have higher amplitude and lower frequency firing functionally correspond to our RNA-seq results indicating upregulation of genes in this line involved in synaptogenesis and calcium signaling.

Surprisingly, IPA-based pathway overrepresentation also found downregulation of pathways involved in p53 signaling, cell cycle checkpoint control, and DNA damage checkpoint regulation in both HET and HOM clones compared to WT (Fig 5A). *TP53* itself was strongly downregulated in both HET and HOM, along with *CHEK2*, *CDK2*, *FAS*, *CDKN1A*, and other genes involved in response to DNA damage and cell death (Fig 5B and 5C, S14 and S15 Figs). This downregulation could be driven by gene expression changes in upstream regulators *NUPR1*, *TRIB3*, *and CDK19* (S13C Fig). The Unfolded Protein Response pathway was also downregulated in rs148726219-edited clones compared to WT; changes in *LONP1*, *ATF3*, *ATF4*, and *CEBPB* expression were detected, all of which are mediators of protein turnover and responses to cellular stress (S13C and S15 Figs). These findings were also replicated in Cnetplot (S16 and S17 Figs). Taken together, these results highlight the phenotypic consequences of rs148726219 in mature iN biology and suggest the role of this SNP in several AD-relevant targets, such as neuron development, signaling, and survival.

## RNA-seq detects two DEGs impacted by rs148726216 throughout hiPSC-iN differentiation

A striking finding from the RNA-seq data was the significant differential regulation of 2 genes —*CAT* and *PCDHB5*—between HET and HOM rs148726219 genotypes and the WT line at all 5 timepoints (Figs 4A, 6A and 6B). Quantification of *CAT* and *PCDHB5* transcripts by qPCR

from 3 independent hiPSC-iN differentiations confirmed that both genes are highest in BIONi010-C-13 parental and WT lines, have moderate expression levels in HET clones, and lower expression levels in HOM clones (S10E and S10F Fig), suggestive of a unique dosage effect of this SNP on expression of these genes. *CAT* encodes catalase, an antioxidant that converts peroxides into water [37], while *PCDHB5* encodes a member of the protocadherin family that are thought to be important for neuronal development [38]. Western blots for catalase confirmed our RNA-seq/qPCR results, which showed catalase protein expression was significantly downregulated in HET and HOM clones compared to both parental and WT lines (Fig 6C, S19 Fig). Whole-genome sequencing of all the cell lines in the study revealed no unintended off-target CRISPR editing or presence of *de novo* SNPs in the *CAT* or *PCDHB5* regions that could explain this result (S20 Fig, S3 Table). Although the mechanism behind this finding is currently unexplained, the robustness of the rs148726219-driven effect is noteworthy.

## Discussion

Although the number of GWAS-identified variants associated with Alzheimer's disease have skyrocketed, very little is known about the mechanisms by which most of these may impact disease. Using Finnish population-based PheWAS, our study nominated an AD-associated SNP tightly linked to a causal *APOE* variant but with potential for an additive effect on LOAD risk in the Finnish population. To functionally characterize rs148726219, we engineered heterozygous and homozygous alleles into hiPSCs, and investigated molecular consequences of this SNP throughout multiple hiPSC-to-neuron differentiation stages. Collectively, we observed that rs148726219-harboring lines develop into early-stage cortical neurons equally as well as WT lines, but their transcriptomes diverge from the WT line almost exclusively in mature iNs. Our data strongly suggest that expression of nearby genes are affected and important signaling pathways relevant to AD pathogenesis are dysregulated in the presence of the risk allele in a mature neuronal context, corroborative of its role as a risk allele for LOAD in the Finnish population. To our knowledge, this study represents one of the only functionalization experiments of a non-coding SNP identified from a large population cohort in an hiPSC-derived neuron model to date. This investigation provides a framework for future use of hiPSC models in the validation and functionalization of non-coding SNPs and their associated disease-modifying genes which could be further developed as novel drug targets. Moreover, our large, well-controlled, and hypothesis-generating RNA-seq dataset provides crucial insights into iN transcriptomes and can spur further investigations into the genomic context of rs148726219 and its role as a risk factor for complex phenotypes such as AD and dementia.

Global transcriptomes of HET and HOM clones compared to WT and parental lines revealed a distinct observation: most genes were only differentially expressed at day 23 of hiPSC-derived cortical neuron differentiation, suggesting that the impact of rs148726219 is specific to mature iN cells. Very few DEGs were identified at earlier differentiation timepoints when cells were transitioning from hiPSCs, neural progenitors, or immature neuron developmental stages. Since AD fundamentally affects neurons later in life, and because rs148726219 was identified in a PheWAS of LOAD patients, these results lend support to the hypothesis that rs148726219 contributes to late onset AD risk. Our NGN2 iN model is a widely used differentiation method yielding excitatory neurons that has previously been applied for drug screening and AD modeling [39]. However, as with any *in vitro* model of CNS disorders, NGN2 iNs have their limitations; specifically: forced differentiation may not fully recapitulate normal development as well as alternative differentiation methods and secondly, the observed expression, in some reports, of peripheral nervous system markers suggest a less well-defined cell identify in iNGN2 iNs [40, 41]. Future studies should apply additional protracted

differentiation methods that do not rely on forced NGN2 overexpression, such as dual SMAD-inhibition-based [42], where neurons develop along a more natural route, or organoid models [43], which better recapitulate neuron-glia interactions found *in vivo*. Additional WT clones may also reveal even more subtle differences between HET, HOM and WT genotypes along the differentiation continuum from neural progenitors to mature neurons.

Several gene networks related to cortical neuron function emerged as differentially regulated in the HET and HOM lines compared to the isogenic WT control in our RNA-seq data. At a pathway level, calcium and CREB signaling, which are important components of synaptogenesis and synaptic transmission, were upregulated in HET and HOM iNs. However, we did not observe differences in hiPSC-iN differentiation potential or cell morphology between genotypes. We suspect these gene expression changes produce subtle effects in neuronal function rather than profound alterations. Activation of these pathways could suggest neuronal hyper-excitability, a pathway that has been linked to epilepsy, has been suggested as a mechanism in AD, and has been observed previously in hiPSC AD models [44]. Indeed, calcium imaging of WT, HET, and HOM lines shows detectable differences in synapse and neuronal circuit maturation between genotypes in day 45 iN cells. The HET-2D2 line shows significantly increased firing amplitude and decreased frequency compared to the WT-2A1 clone, indicating a more mature and excitable functional neuronal network. Indeed, there is recent evidence for neuron hyperexcitability in AD neurons and in hiPSC-derived models of the brain [44, 45]. While the HOM-2H6 line does not show significant differences from WT-2A1, it also trends towards increased firing amplitude and decreased frequency. These functional outcomes correspond to top upregulated DEGs involved in calcium signaling and synaptogenesis. However, functional differences between HOM and HET lines are not explained by their transcriptomic outcome since both lines showed similar levels of DEG upregulation. These results may arise from differences in timing of RNA-seq (day 23 of iN maturation) and calcium imaging (day 45 of iN maturation); future studies should investigate gene expression differences on day 45 to determine whether more distinction between HET and HOM genotypes appear as neuronal circuits further mature. Altogether, these late-stage expression patterns are congruent with onset of AD-related molecular changes in mature neurons. To relate rs148726219 to AD pathogenesis in neuronal function, more extensive functional evaluation of rs148726219 effects should be completed in iNs (for example: more engineered lines tested by MEA and calcium imaging as well as evaluation of pathologic amyloid/tau seeding to connect upregulated circuit function to AD pathology) [46]. Additionally, it is possible that other cell types of the brain, such as microglia and astrocytes, are also influenced by rs148726219 and warrants further exploration. It also remains to be probed if these changes are reproducible in three-dimensional or *in vivo* models where glial cells may play critical mediating roles.

Additional changes to pathways involved in cell death, unfolded protein response (UPR) and apoptosis gene sets suggest differential but subtle responses to cellular stress between the HET, HOM and WT lines. Downregulation of *ATF3/4* in HET and HOM lines suggests decreased responses to metabolic/ER stress. ATF4 is known to be dysregulated in neurodegenerative disorders and is a potential target for therapeutics [47]. Moreover, downregulation of cell cycle/p53/apoptosis offers a mechanism for cells to compensate for downregulated UPR stress in order to maintain cell survival. As we did not observe changes in cell viability or cell numbers between genotypes, it is unclear whether these molecular changes translate to physiological changes in cell survival or cellular senescence.

Our expectation that *cis* effects of rs148726218 would mediate gene expression changes in *FOSB* and its splice isoform, *ΔFOSB*, in iN cells was not observed. Moreover, *FOSB* is lowly expressed in healthy cortical neurons, which is corroborated by brain expression data from GTEx [30], potentially downplaying its role in neuronal cell types. However, western blots and

our RNA-seq analysis revealed that expression of ΔFOSB, *ERCC1*, and a handful of other genes within 1 Mb of rs148726219 were subtly dysregulated by the presence of this SNP specifically in day 23 iN cells, suggesting that rs148726219 can modulate the expression of nearby genes. Although GTeX does not implicate an eQTL role for rs148726219 on nearby genes in brain cell types, other databases suggest a putative eQTL interaction with rs148726219 in alternative tissue types (e.g., muscle and adipose; https://fivex.sph.umich.edu/about). Our observations highlight a weak but statistically significant eQTL effect for rs148726219 and suggest 2 interpretations: 1) that this SNP may exert an additive effect to *APOE* causal variants within the LD block, and/or 2) that rs148726219 exerts its strongest eQTL effects in other human cell types unrelated to cell types of the brain.

Simultaneously, it remains possible that the SNP may act through *trans*-chromosomal mechanisms that we have not yet deciphered to regulate genes such as *CAT* and *PCDHB5*, observed in our datasets. Previous reports have identified *trans*-eQTLs for AD and AD pathological hallmarks in multiple genes, including *APOC1* on chromosome 19 [48]. Alternatively, rs148726219 may act in concert with other SNPs in the same region or at longer distances on chr19 to exert a functional effect in neurons [26]. We found it interesting that *CAT* and *PCDHB5* expression patterns decreased in a genotype-dependent fashion with rs148726219 and independent of differentiation stage, though both are located on different chromosomes from the SNP. We did find that known regulators of *CAT*, such as *XBP1* and *CEBPB*, were dysregulated in our RNA-seq dataset (S21 Fig), suggesting that potential *trans* mechanisms mediated by rs148726219 may exist. Catalase is a highly conserved enzyme that converts peroxides to water and oxygen and protects cells from oxidative damage. Reduced antioxidant capability and mitochondrial dysfunction are known components of AD [49, 50]; however, whether these are causative or simply byproducts of disease remains unclear. The protocadherin gene family is made up of clustered α, β, and γ *PCDH* genes, which are primarily expressed in the central nervous system and mediate important cell-cell signaling cascades. Due to the diversity and combinatorial expression patterns of protocadherins (~53 neuronal genes), they have been proposed to function as synaptic "barcodes", with each cell in the brain expressing its own unique combination [51, 52]. Protocadherins have been implicated in several neurological disorders, including Alzheimer's, as they are processed by γ-secretase [53]. These potential impacts of *CAT* and *PCDHB5* in AD biology and their connections to rs148726219 warrants further investigation.

In conclusion, this study establishes hiPSC-derived cortical neurons as a relevant two-dimensional model of AD pathophysiology and provides experimental fine-mapping of a complex AD-associated region. Though we begin to characterize the molecular consequences of rs148726219 on AD risk, much more research is needed to fully understand the mechanism(s) behind the association, particularly in additional functional assays and disease-relevant cell types. However, these findings represent an important initial step towards the identification of novel genes underlying disease etiology and biomarkers that may predict disease development, which will hopefully translate to the clinic. The recent positive results for Aβ-targeting drugs lends encouragement for the transformative potential of additional genetically-validated therapeutic targets.

## Supporting information

**S1 Fig. Finngen disease associations for rs148726219.** Finngen metadata (data freeze 3) for rs148726219 showing associations with Alzheimer's disease and dementia.
(PDF)

**S2 Fig. Cytogenetic analysis of rs148726219-edited hiPSC clones.** G-banding was performed on 20 metaphase cells. All lines exhibited normal male karyotypes. **A.** WT-2A1. **B.** Het-2D2. **C.** Het-2G6. **D.** Hom-2B11. **E.** Hom-2H6.
(PDF)

**S3 Fig. Day 23 iNeurons do not show differences between clones in cell morphology, levels of neuronal marker proteins, or viability. A.** Representative immunofluorescence images of CUX1 (green), SOX2 (orange), and TUJ1 (red) are shown for the BIONi010-C-13 parental line, and WT-2A1, HET-2D2, HET-2G6, HOM-2B11, and HOM-2H6 clones. The merged images includes DAPI nuclear counterstain (blue). Scale bars = 100 um. **B.** Relative cell viability of each iN line at day 23 measured by Cell Titer Glo. Relative fluorescence units from 3 independent differentiation experiments, each with 6 technical replicates, was normalized to the WT-parental line (n = 3, n = 2 for HOM-2H6). ns = not significant. *p < 0.05 by one-way ANOVA with multiple comparisons.
(PDF)

**S4 Fig. High-throughput qPCR data from rs148726219-edited lines. A.** Principal component analysis (PCA) of high-throughput qPCR of BIONi010-C-13 parental, WT-2A1, HET-2D2, HET-2G6, HOM-2B11, and HOM-2H6 lines at days 0, 2, 6, 13, and 23 of 1 representative iNeuron differentiation. Three technical replicates of each line are shown per timepoint. Samples cluster by timepoint, not by genotype or clone. NTC = qPCR no template control; UHRR = universal human reference RNA used as a positive control for gene expression. **B-C.** Neurogenin2 (*NEUROG2*, *NGN2*) expression measured by high-throughput qPCR as in (A). Values were normalized relative to the geometric mean of 3 housekeeping genes (*GAPDH*, *EIF4A2*, *RPL13*) and expressed as fold change values (DCt). Data grouped by timepoint (**B**) and by line (**C**) illustrates increased *NEUROG2* expression with DOX induction (days 2 and 6), and reduced expression following treatment (days 13 and 23). No expression is detected at day 0, indicating no inherent leakiness of the Tet operator.
(PDF)

**S5 Fig. High-throughput qPCR data of marker genes from rs148726219-edited lines.** Expression of genes related to **A.** mature neurons (*DCX*, *DLG4*, *MAP2*, *MAPT*, *RBFOX3*, *SLC17A6*, *SLC17A7*, *SYN1*, *SYP*, *TUBB3)*, **B.** pluripotency (*LIN28A*, *MKI67*, *NANOG*, *POU5F1)*, **C.** neural progenitor cells (*DACH1*, *OTX2*, *PAX6*, *PROM1*, *SOX2*), **D.** non-neuronal cells *(CDX2*, *GATA4*, *GFAP*, *OLIG1*, *S100B*, *SLC1A3*, *SOX17*, *TBXT)*, and **E.** AD-associated markers (*APOE*, *4R* isoform of *MAPT*, *PSEN1*, *APP)*. Transcripts were measured by high-throughput qPCR of BIONi010-C-13 parental, WT-2A1, HET-2D2, HET-2G6, HOM-2B11, and HOM-2H6 lines at days 0, 2, 6, 13, and 23 of 1 representative iN differentiation. Values were normalized relative to the geometric mean of 3 housekeeping genes (*GAPDH*, *EIF4A2*, *RPL13*) and expressed as fold change values. Three technical replicates of each line are shown per timepoint.
(PDF)

**S6 Fig. Secreted levels of amyloid beta from rs148726219-edited lines.** Expression of Aβ protein isoforms Aβ40 (**A**), Aβ38 (**B**), and Aβ42 (**C**) measured by MSD ELISA from media conditioned by BIONi010-C-13 parental, WT-2A1, HET-2D2, HET-2G6, HOM-2B11, and HOM-2H6 lines on day 23 of iN differentiation. The ratio of Aβ42/Aβ40 is also shown (**D**). Cells were untreated (blue bars), treated with vehicle control (DMSO; grey bars), or treated with DAPT to inhibit Aβ production (white bars) for 24 hrs prior to media collection. Values were normalized against a standard curve and expressed relative to total cell number, quantified as relative fluorescence units from Cell Titer Glo assays. n = 3 independent hiPSC-

iNeuron differentiations (or n = 2 for HOM-2H6), with each represented by a black dot. No apparent differences in Aβ levels are observed between lines with heterozygous/homozygous alleles for rs148726219 and wild type lines.
(PDF)

**S7 Fig. Principle component analysis plots of RNA-seq data.** PCA plots are colored by time point (**A**), colored by rs148726219-edited clone (**B**), and colored by BIONi010-C-13 parental line versus WT-2A1 clone (**C**). Samples cluster by time point, not by line. No apparent differences are detectable between the parental line and WT clone.
(PDF)

**S8 Fig. Volcano plots of RNA-seq data from homozygous clones versus wild type.** Homozygous clones (HOM-2B11, HOM-2H6) were compared to the WT clone (WT-2A1) at days 0, 2, 6, 13, and 23 of hiPSC–iNeuron differentiation. Each gene is represented by a dot showing– log10(adjusted p-value) (ie. FDR) and log(fold-change) (logFC) values. Downregulated genes in blue, upregulated genes in red.
(PDF)

**S9 Fig. Volcano plots of RNA-seq data from heterozygous clones versus wild type.** Heterozygous clones (HET-2D2, HET-2G6) were compared to the WT clone (WT-2A1) at days 0, 2, 6, 13, and 23 of hiPSC–iNeuron differentiation. Each gene is represented by a dot showing– log10(adjusted p-value) (ie. FDR) and log(fold-change) (logFC) values. Downregulated genes in blue, upregulated genes in red.
(PDF)

**S10 Fig. Gene expression measured by SYBR Green RT-qPCR.** Values for BIONi010-C-13 WT-parental line and WT-2A1 clone (grey bars), rs148726219-heterozygous clones (blue bars), and homozygous clones (green bars) day 0, 2, 6, 13, and 23 of hiPSC-iNeuron differentiation are shown. Cq values were normalized to the geometric mean of 3 housekeeping genes (*B2M*, *ACTB*, *GAPDH*) and where possible, are expressed relative to the WT-parental line. **A.** Total *FOSB* transcript. Day 0, n = 3; Day 2, n = 2; Day 23, n = 4. **B.** D*FOSB*. Day 0, n = 3; Day 2, n = 2; Day 23, n = 4. **C.** Total *ERCC1* transcript. Day 0, n = 3; Day 2, n = 2; Day 23, n = 4. **D.** *ERCC1* long transcript isoform. Day 0, n = 3; Day 2, n = 2; Day 23, n = 4. **E.** *CAT*. n = 2. **F.** *PCDHB5*. n = 2.
(PDF)

**S11 Fig. Western blot for FOSB expression. A.** Western blot of day 23 iNeuron lysates from BIONi010-C-13 parental, WT-2A1, HET-2D2, HET-2G6, HOM- 2B11, and HOM-2H6 lines either untreated (-) or stimulated for 4 hrs with PMA (+). Blot was probed with an antibody against FOSB, which detects canonical FOSB (top band, ~48 kD) and DFOSB (bottom band, ~38 kD). GAPDH is shown as the loading control. **B.** Quantification of the blot in (A) by pixel densitometry for canonical FOSB (dark grey area) and DFOSB (light grey area) relative to GAPDH. The height of each bar represents the total amount of FOSB (canonical FOSB + DFOSB). n = 1.
(PDF)

**S12 Fig. Expression of genes within 1 Mb of rs148726219.** Expression values are expressed as fragments per kilobase per million mapped reads (FPKM) for WT-2A1 (green), rs148726219-heterozygous (HET-2D2, HET-2G6; orange), and homozygous (HOM-2B11, HOM-2H6; purple) clones at day 0, 2, 6, 13, and 23 of hiPSC-iNeuron differentiation. Four technical replicates for each clone are shown with each represented by a black dot. **A.** *ERCC1*. **B.** *BCL3*. **C.** PPP1R37. **D.** *RTN2*. **E.** *DMPK*. **F.** *APOE*.
(PDF)

**S13 Fig. IPA gene ontology analysis showing dysregulated pathways and upstream regulators in day 23 iNeurons. A., B.** Top 10 dysregulated pathways in homozygous clones (HOM-2B11, HOM-2H6; A) and heterozygous clones (HET-2D2, HET-2G6; B) compared to WT-2A1 clone.–Log(p-value) denotes level of significance for each pathway. Z-scores are denoted by numbers above each bar and colors (upregulated = orange, downregulated = navy). **C.** Top 20 dysregulated upstream regulators from comparison analysis between homozygous clones (left column) and heterozygous clones (right column) relative to WT-2A1 clones on day 23. Z-scores denote level of upregulation (orange) or downregulation (navy).
(PDF)

**S14 Fig. IPA gene ontology analysis showing dysregulated genes within selected pathways at day 23.** Differentially expressed genes (based on FDR < 0.05, logFC < -1.0 and > 1.0) are shown as heatmaps for each pathway. Comparison analysis between homozygous clones (left column) and heterozygous clones (right column) relative to WT-2A1 clones on day 23. Z-scores denote level of upregulation (orange) or downregulation (navy). Color denotes log fold change (logFC) values; orange = upregulated, navy = downregulated.
(PDF)

**S15 Fig. Expression of genes driving pathway dysregulation in cells with edited rs148726219 alleles.** Expression values are expressed as fragments per kilobase per million mapped reads (FPKM) for WT-2A1 (green), rs148726219-heterozygous (HET-2D2, HET-2G6; orange), and homozygous (HOM-2B11, HOM-2H6; purple) clones at day 0, 2, 6, 13, and 23 of hiPSC-iNeuron differentiation. Four technical replicates for each clone are shown with each represented by a black dot.
(PDF)

**S16 Fig. Cnetplots of rs148726219-homozygous iNeurons compared to the WT clone on day 23.** Size of each node represents level of pathway enrichment. Fold change represents logFC values of each gene compared to the WT. **A.** Downregulated pathways. **B.** Upregulated pathways.
(PDF)

**S17 Fig. Cnetplots of rs148726219-heterozygous iNeurons compared to the WT clone on day 23.** Size of each node represents level of pathway enrichment. Fold change represents logFC values of each gene compared to the WT. **A.** Downregulated pathways. **B.** Upregulated pathways.
(PDF)

**S18 Fig. Calcium signaling imaging of iN cells.** (top) Single-frame images of day 45 iNs (WT 2A1, HOM 2H6, HET 2D2) transduced with AAV6-syn-GCaMP6f. (bottom) Representative raster plots visualizing neuronal circuit activity of each line. Individual lines show firing events of 50 iN cells per frame, imaged over 2000 frames (1 sec/frame) and lineup of individual lines indicates neuronal circuit formation. rs148726219-edited heterozygous hiPSC line (HET-2D2) shows enhanced functional circuit maturation compared to wildtype (WT-2A1) and homozygous (HOM-2H6) lines.
(PDF)

**S19 Fig. Additional western blots for catalase expression used to perform quantification.** Additional western blots of day 23 iNeuron lysates from independent hiPSC-iN differentiations of BIONi010-C-13 parental, WT-2A1, HET-2D2, HET-2G6, HOM-2B11, and HOM-2H6 clones. Blots were probed with an antibody against catalase and GAPDH as a loading control. The first experimental replicate is shown in Fig 6. **A.** Second experimental replicate. **B.**

Third experimental replicate.
(PDF)

**S20 Fig. WGS analysis of *CAT* and *PCDHB5* in rs148726219-edited hiPSC lines.** Reads were aligned to hg38 using VarSeq. BIONi010-C-13 WT-parental, WT-2A1, HET-2D2, HET-2G6, HOM-2B11, and HOM-2H6 lines are shown as separate rows. Detected variants are shown as colored lines. No variants were identified in coding regions. Furthermore, no variants were detected that would explain *CAT* (top) or *PCDHB5* (bottom) decreases in heterozygous and homozygous clones compared to parental or WT-2A1 clones.
(PDF)

**S21 Fig. Expression of known regulators of catalase and members of the AP-1 complex identified from RNA-seq on day 23 of hiPSC-iNeuron differentiation.** Log fold change (logFC) and FDR (adj.P.Val) values are shown for each gene in heterozygous clones (HET-2D2, HET-2G6) and homozygous clones (HOM-2B11, HOM-2H6) compared to the wild type clone (WT-2A1). Significant genes are denoted by colored boxes (blue = downregulated, orange = upregulated). "-" denotes genes not detected by RNA-seq. None of these genes were differentially expressed at any other time point.
(PDF)

**S1 Table. Fluidigm delta gene assay panel.**
(XLSX)

**S2 Table. Differentially expressed genes within 1 Mb of *FOSB*.**
(XLSX)

**S3 Table. Variants identified by whole genome sequencing for *CAT* and *PCDHB5* loci.**
(XLSX)

**S1 Movie. Calcium signaling imaging of HET-2D2 iNeurons.**
(MOV)

**S2 Movie. Calcium signaling imaging of WT-2A1 iNeurons.**
(MOV)

**S3 Movie. Calcium signaling imaging of HOM-2H6 iNeurons.**
(MOV)

## Acknowledgments

We want to acknowledge the participants and investigators of the FinnGen study. Thank you to AbbVie employees Areej Ammar for performing whole genome sequencing, Bridget Riley-Gillis for conditional GWAS and Victor Coll for manuscript figure organization. Special thanks to Allison Ebert (Medical College of Wisconsin, no funding to disclose) for consulting on valuable scientific discussions.

## Author Contributions

**Conceptualization:** Lindsay R. Stolzenburg, Sahar Esmaeeli, Ameya S. Kulkarni, Taekyung Kwon, Christina Preiss, Joshua D. Stender, Justine D. Manos, Peter Reinhardt, Jeffrey F. Waring, Cyril Y. Ramathal.

**Data curation:** Lindsay R. Stolzenburg, Ameya S. Kulkarni, Cyril Y. Ramathal.

**Formal analysis:** Lindsay R. Stolzenburg, Sahar Esmaeeli, Ameya S. Kulkarni, Fedik Rahimov.

**Investigation:** Lindsay R. Stolzenburg, Sahar Esmaeeli, Ameya S. Kulkarni, Erin Murphy, Taekyung Kwon, Christina Preiss, Joshua D. Stender, Peter Reinhardt, Fedik Rahimov.

**Methodology:** Lindsay R. Stolzenburg, Sahar Esmaeeli, Ameya S. Kulkarni, Taekyung Kwon, Lamiaa Bahnassawy, Joshua D. Stender, Justine D. Manos, Peter Reinhardt, Fedik Rahimov, Jeffrey F. Waring, Cyril Y. Ramathal.

**Project administration:** Peter Reinhardt, Jeffrey F. Waring.

**Resources:** Lindsay R. Stolzenburg, Lamiaa Bahnassawy, Joshua D. Stender, Justine D. Manos, Peter Reinhardt, Fedik Rahimov, Jeffrey F. Waring.

**Software:** Lindsay R. Stolzenburg, Ameya S. Kulkarni.

**Supervision:** Taekyung Kwon, Justine D. Manos, Jeffrey F. Waring, Cyril Y. Ramathal.

**Validation:** Lindsay R. Stolzenburg, Ameya S. Kulkarni, Justine D. Manos.

**Visualization:** Lindsay R. Stolzenburg, Sahar Esmaeeli, Ameya S. Kulkarni, Cyril Y. Ramathal.

**Writing – original draft:** Lindsay R. Stolzenburg, Taekyung Kwon, Lamiaa Bahnassawy, Joshua D. Stender, Justine D. Manos, Peter Reinhardt, Fedik Rahimov, Jeffrey F. Waring, Cyril Y. Ramathal.

**Writing – review & editing:** Lindsay R. Stolzenburg, Sahar Esmaeeli, Erin Murphy, Taekyung Kwon, Christina Preiss, Lamiaa Bahnassawy, Joshua D. Stender, Justine D. Manos, Peter Reinhardt, Fedik Rahimov, Jeffrey F. Waring, Cyril Y. Ramathal.

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
