## [Decision Letter · Decision Letter 0]

14 Jun 2023

PONE-D-23-13309Functional characterization of a single nucleotide polymorphism associated with Alzheimer’s Disease in a hiPSC-based neuron model.PLOS ONE

Dear Dr. Ramathal,

Thank you for submitting your manuscript to PLOS ONE. After careful consideration, we feel that it has merit but does not fully meet PLOS ONE’s publication criteria as it currently stands. Therefore, we invite you to submit a revised version of the manuscript that addresses the points raised during the review process.

We look forward to receiving your revised manuscript.

Kind regards,

Amy McCart Reed

Academic Editor

PLOS ONE

Journal Requirements:

Additional Editor Comments:

Thank you for your submission.

In addition to addressing the points as noted by the reviewers, could you provide further clarity around the restrictions for data sharing?

Reviewers' comments:

Reviewer's Responses to Questions

**Comments to the Author**

1. Is the manuscript technically sound, and do the data support the conclusions?

Reviewer #1: Partly

Reviewer #2: Yes

2. Has the statistical analysis been performed appropriately and rigorously? 

Reviewer #1: Yes

Reviewer #2: No

3. Have the authors made all data underlying the findings in their manuscript fully available?

Reviewer #1: Yes

Reviewer #2: No

4. Is the manuscript presented in an intelligible fashion and written in standard English?

Reviewer #1: Yes

Reviewer #2: Yes

5. Review Comments to the Author

Reviewer #1: The study proposed provides a characterization of a rare AD-linked variant by using hiPSC-derived neuronal cultures, CRISPR-Cas9, RNA-seq and imaging analyses.

Overall, the rationale of the work is clear and the article is well written.

Some points need to be clarified.

In general, the resolution of some images is too low, making it difficult to read the data.

• Row 350: the authors declare that generated hiPSC have normal karyotype when compared to the parental line but in Supplemental Fig.3, qPCR-based karyotyping shows for chr18q a difference in HET lines.

• Row 371 and 667-668: the authors declare that no impairment of cell viability is detected on differentiated cultures. This conclusion is based on qualitative observations of differentiated cells. The use of more specific markers for apoptosis (i.e., cleaved Caspase 3 immunofluorescence) could sustain this observation. The authors declare that no differences in cell number were observed. Did they quantify the number of differentiated cells?

• Row 456-463: Authors declare that no differences in FOSB expression was detected also due to low CT values allowing for reliable quantification. Did the authors quantify the FOSB WB in Supplemental Figure 18? It would be helpful for identifying subtle differences.

• Row 636: the authors declare that no differences in synapses morphology were observed. How did they investigate this point?

• Rows 543-553: The impairment in neuronal firing is mostly evident in heterozygous lines, even if they do not differ in terms of DEGs with homozygous. Can the authors comment more in detail this result?

• Figure 1D. There is an overlap between the brackets for day 23 and 13.

• Figure 5A. Color of terms need to be adjusted according to the background color.

• Figure 5D. The WT line in the column chart is named hFOSWT, which never appeared in the text.

Reviewer #2: In this paper Stolzenburg at al characterize induced neurons (iNs) generated from iPSCs that have been modified by CRISPR-Cas to be homozygote or heterozygote carriers of rs148726219 SNP or WT controls. The authors present an extensive set of data of the cell gene expression, functionality and protein expression and conclude that while the studied SNP does not alter iN differentiation it affects the cell phenotype in terms of functionality as well as gene and protein expression levels. The manuscript is well written and an extensive amount of supporting data is presented. Therefore I support the acceptance of this manuscript, however, have some minor comments that the authors should address.

1. This study utilizes human cell lines, however authors state that ethics were not needed. Can authors clarify on this?

2. Data availability: Authors state that not all data is available (that restrictions apply), but this is not further clarified

3. The manuscript contains and extensive amount of supplementary material (in total 28 supplementary figures). The authors should carefully consider if all main data is presented in the main manuscript and if all supplementary data is necessary for the storyline of the manuscript. If results presented is supplementary material is not discussed in the main text, they should be omitted from the paper (examples include: the authors don’t mention the results for NEUROG2 and MBP, some of the pluripotency markers and AD markers although they are shown in the supplementary figures). All results shown in main and supplementary figures should be at least briefly touched upon in the text or alternatively removed from the figures.

4. Are authors able to clarify the origin of the used iPSC line (that is not of Finnish origin)?

5. Statistics: authors should clarify the number of independent and technical replicates used for experiments in the study. In addition, authors mention that Student’s t-test was used, however, this is not an appropriate statistical test when more than 2 groups are compared. Authors should carefully review statistical tests used.

6. The authors should clarify further the use of the two selected timepoints: why was D23 used for gene expression comparison and D45 for calcium signaling assays? Why wasn’t the same timepoint used for both assays? In addition, from line 489 onwards: authors state that hundreds of DEGs were identified between HET/HOM and WT at D23 when iN are most mature. However, this wasn’t the “most mature” state of the cells since later it’s concluded that the cells were mature at D45. Authors should correct wording in this sentence. Authors should also discuss more thoroughly the limitations of comparing these two timepoints for gene expression/functional activity.

7. There are some abbreviations throughout the manuscript that the authors should define:

• In the abstract: ERCC1, FOSB

• Methods: DOX NGN2 (these are later defined in the results, but should be defined when first mentioned), NGS

• Results: ECM

8. Discussion: The authors mention studies like this could be used for novel drug target identification – could the authors elaborate this more with a sentence or two.

9. Line 626: authors mention that NGN2 iNs have their limitations – what are these limitations?

10. Discussion should include more discussion on the impact of the findings and potential translation into the clinic

6. PLOS authors have the option to publish the peer review history of their article (what does this mean?). If published, this will include your full peer review and any attached files.

Reviewer #1: No

Reviewer #2: No

---

## [Author Response · Author response to Decision Letter 0]

31 Jul 2023

Dear Editor, July 31, 2023

Thank you for the opportunity to revise our manuscript entitled “Functional characterization of a single nucleotide polymorphism associated with Alzheimer’s Disease in a hiPSC-based neuron model” for consideration in PLoS One. We have carefully reviewed the Editor’s and Reviewer comments made and made the appropriate revisions or clarifications in the manuscript documents, as well as addressed them in the letter below. Regarding data sharing, We have uploaded our entire RNA sequencing dataset into GEO and this will be accessible via a private link and password until/if the manuscript is accepted for publication in PLoS One. The link to the data upload on GEO is accessible at: https://www.ncbi.nlm.nih.gov/geo/query/acc.cgi?acc=GSE239367 . Enter token mrsnqogmxzcrfqn into the box

Please see our responses in blue font below point-by-point for each comment.

Additional Editor Comments:

Thank you for your submission.

In addition to addressing the points as noted by the reviewers, could you provide further clarity around the restrictions for data sharing? There are no restrictions, and the data has been uploaded to GEO and is currently in private view in GEO. We have uploaded our entire RNA sequencing dataset into GEO and this will be accessible via a private link and password until/if the manuscript is accepted for publication in PLoS One. The link to the data upload on GEO is accessible at: https://www.ncbi.nlm.nih.gov/geo/query/acc.cgi?acc=GSE239367. Enter token mrsnqogmxzcrfqn into the box

Responses to Reviewer 1:

Reviewer #1: The study proposed provides a characterization of a rare AD-linked variant by using hiPSC-derived neuronal cultures, CRISPR-Cas9, RNA-seq and imaging analyses.

Overall, the rationale of the work is clear and the article is well written.

Some points need to be clarified.

• In general, the resolution of some images is too low, making it difficult to read the data. We have generated new PDFs of the figures at a higher resolution and hope this will solve the problem.

• Row 350: the authors declare that generated hiPSC have normal karyotype when compared to the parental line but in Supplemental Fig.3, qPCR-based karyotyping shows for chr18q a difference in HET lines. qPCR-based karyotyping was performed for each line over the course of 4 separate experiments (chr18q data from Supplemental Figure 3 is broken out into each individual experiment below). In Exp2, the WT-parental line shows an apparent decrease in copy number compared to the standardized control provided with the kit, but this was not reflected in Exp1 or Exp3. The most likely source of this difference is due to inherent variability commonly observed between qPCR-based experiments. Importantly, in the data below, none of the edited lines show differences in chr18q copy number compared to the WT-parental line. Moreover, G-banding and STR analyses did not identify any abnormal karyotypes in any line. Because of the qPCR inter-experimental variability, and in an effort to reduce the number of redundant Supplemental Figures (see point #3 in response to Reviewer #2), we have opted to remove Supplemental Figure 3 from the manuscript. The text has been updated accordingly (lines 128-131, 356-357).

• Row 371 and 667-668: the authors declare that no impairment of cell viability is detected on differentiated cultures. This conclusion is based on qualitative observations of differentiated cells. The use of more specific markers for apoptosis (i.e., cleaved Caspase 3 immunofluorescence) could sustain this observation. The authors declare that no differences in cell number were observed. Did they quantify the number of differentiated cells? We used Cell Titer Glo to quantify the number of viable cells on Day 23 of differentiation across 3 independent experiments. With the exception of a slight increase in the number of HOM-2G6 cells (~16% compared to the parental line), there were no other significant differences in viability between lines. This data, along with corresponding text (lines 382-384), has been added as Supplemental Figure 3B.

• Row 456-463: Authors declare that no differences in FOSB expression was detected also due to low CT values allowing for reliable quantification. Did the authors quantify the FOSB WB in Supplemental Figure 18? It would be helpful for identifying subtle differences. The WB was quantified (now Supplemental Figure 11B). Following PMA treatment, there appears to be a trend toward increased total FOSB expression in all 5 clones compared to the WT-parental line, but no obvious differences in total FOSB were detected between the rs148726219-edited lines and the WT clone. However, in the absence of PMA, it does appear that the HOM clones trend toward slightly increased baseline expression of �FOSB compared to the other lines. This analysis only represents 1 experiment and is therefore not statistically significant, so we cannot draw concrete conclusions based on it. We have modified the text to indicate these findings (lines 480-483, 724, 727-728).

• Row 636: the authors declare that no differences in synapses morphology were observed. How did they investigate this point? Beyond calcium imaging to measure synapse connectivity, we did not specifically investigate markers of synapses to determine their morphology. The statement was modified to “cell morphology” instead of “synapse morphology” (line 688) to better harmonize with statements made throughout the rest of the manuscript.

• Rows 543-553: The impairment in neuronal firing is mostly evident in heterozygous lines, even if they do not differ in terms of DEGs with homozygous. Can the authors comment more in detail this result? Gene expression analyses were completed on Day 23 of differentiation while calcium imaging was performed on Day 45. We have modified the text to clarify that Day 23 of differentiation represents an early iN maturation timepoint (lines 511-512), and that synapses reach further maturity by Day 45 (line 568). Our internal data suggests that full circuit maturation in iN cultures occurs after Day 34; we used this information, together with imaging readouts, to choose Day 45 as the timepoint for this assay. Because calcium imaging and gene expression analyses were not completed at the same timepoint, this could account for the incongruent results between Day 23 and Day 45 that this reviewer mentions. We have added a statement to clarify that additional distinction between the HET and HOM lines could emerge between Day 23 and 45 and that future studies should investigate this point (lines 703-705).

• Figure 1D. There is an overlap between the brackets for day 23 and 13. We assume this is referring to Figure 3D. The overlap is correct – the hierarchical clustering algorithm clustered the data for HET-2D2_Day13_rep1 and HOM-2H6_Day13_rep3 as overlapping with the data from Day 23. The brackets have been modified to visually clarify this. We have also added a description of this in figure legend for Fig 1.

• Figure 5A. Color of terms need to be adjusted according to the background color. The text color was adjusted for better readability.

• Figure 5D. The WT line in the column chart is named hFOSWT, which never appeared in the text. The labels were adjusted to align with terminology used throughout the manuscript.

Reviewer #2: In this paper Stolzenburg at al characterize induced neurons (iNs) generated from iPSCs that have been modified by CRISPR-Cas to be homozygote or heterozygote carriers of rs148726219 SNP or WT controls. The authors present an extensive set of data of the cell gene expression, functionality and protein expression and conclude that while the studied SNP does not alter iN differentiation it affects the cell phenotype in terms of functionality as well as gene and protein expression levels. The manuscript is well written and an extensive amount of supporting data is presented. Therefore I support the acceptance of this manuscript, however, have some minor comments that the authors should address.

1. This study utilizes human cell lines, however authors state that ethics were not needed. Can authors clarify on this? The BIONi010-C-13 NGN2 line used in this manuscript (Schmid et al, Stem Cell Research, 2021) was engineered from an iPSC line generated and described previously (Rasmussen et al, Stem Cell Reports, 2014). We obtained BIONi010-C-13 from the EBiSC under an ethical and legal framework that involved completing an Access and Use Agreement. The EBiSC “reviews all aspects of the informed consent used at the time of sample collection to ensure that individuals sharing tissue and data are fully informed that their donation may be used for iPSC generation, characterization and distribution.” 

2. Data availability: Authors state that not all data is available (that restrictions apply), but this is not further clarified Thank you for this suggestion. We have uploaded our entire RNA sequencing dataset into GEO and this will be accessible via a private link and password until/if the manuscript is accepted for publication in PLoS One. The link to the data upload on GEO is accessible at: https://www.ncbi.nlm.nih.gov/geo/query/acc.cgi?acc=GSE239367. Enter token mrsnqogmxzcrfqn into the box

3. The manuscript contains and extensive amount of supplementary material (in total 28 supplementary figures). The authors should carefully consider if all main data is presented in the main manuscript and if all supplementary data is necessary for the storyline of the manuscript. If results presented is supplementary material is not discussed in the main text, they should be omitted from the paper (examples include: the authors don’t mention the results for NEUROG2 and MBP, some of the pluripotency markers and AD markers although they are shown in the supplementary figures). All results shown in main and supplementary figures should be at least briefly touched upon in the text or alternatively removed from the figures. For the high-throughput Fluidigm qPCR data, graphs not mentioned in the text were removed and the remaining graphs were condensed into a single figure (now Supplementary Figure 5). Moreover, additional supplemental figures, listed below, were removed, as they were not specifically mentioned in the text and we deemed them less critical for the storyline of the manuscript:

- Original Supplemental Figure 3: qPCR-based karyotyping (owing to inter-experimental variability -see additional description in 2nd bullet response to Reviewer #1 above)

- Original Supplemental Figure 12: High throughput qPCR housekeeping gene expression (and its corresponding text, lines 428-430)

- Original Supplemental Figure 14: Venn diagrams

- Original Supplemental Figure 22 (now part of Supplemental Figure 15): sub-panels of RNA-seq expression of genes driving pathway dysregulation that were not mentioned in the text

4. Are authors able to clarify the origin of the used iPSC line (that is not of Finnish origin)? The line is derived from a Black/African American male. Statements in the Methods and Results sections were added to clarify the line’s origin (lines 110-111, 351-352).

5. Statistics: authors should clarify the number of independent and technical replicates used for experiments in the study. In addition, authors mention that Student’s t-test was used, however, this is not an appropriate statistical test when more than 2 groups are compared. Authors should carefully review statistical tests used. We thank the reviewer for this suggestion; figure legends have been adjusted to clarify technical and biological (n) replicates used, if not already specified. Any relevant statistical tests were also specified in the corresponding legends. Note that we modified the statistical test used in Figure 6C (one way ANOVA with multiple comparisons), which slightly changed the resulting p-values. A statement in the “Statistics” Methods section was added to clarify that statistical tests used are now described in their corresponding figure legends (lines 278-281).

6. The authors should clarify further the use of the two selected timepoints: why was D23 used for gene expression comparison and D45 for calcium signaling assays? Why wasn’t the same timepoint used for both assays? In addition, from line 489 onwards: authors state that hundreds of DEGs were identified between HET/HOM and WT at D23 when iN are most mature. However, this wasn’t the “most mature” state of the cells since later it’s concluded that the cells were mature at D45. Authors should correct wording in this sentence. Authors should also discuss more thoroughly the limitations of comparing these two timepoints for gene expression/functional activity. Gene expression analyses were completed on Day 23 of differentiation while calcium imaging was performed on Day 45. We have modified the text to clarify that Day 23 of differentiation represents an early iN maturation timepoint (lines 511-512), and that synapses reach further maturity by Day 45 (line 568). Our internal data suggests that full circuit maturation in iN cultures occurs after Day 34; we used this information, together with imaging readouts, to choose Day 45 as the timepoint for this assay. Because calcium imaging and gene expression analyses were not completed at the same timepoint, this could account for the incongruent results between Day 23 and Day 45 that this reviewer mentions. We have added a statement to clarify that additional distinction between the HET and HOM lines could emerge between Day 23 and 45 and that future studies should investigate this point (lines 703-705). 

7. There are some abbreviations throughout the manuscript that the authors should define:

• In the abstract: ERCC1, FOSB

• Methods: DOX NGN2 (these are later defined in the results, but should be defined when first mentioned), NGS

• Results: ECM

Abbreviations have been defined:

- FOSB (line 36)

- ERCC1 (line 36-37)

- DOX (line 108)

- NGN2 (line 109)

- NGS (line 123)

- ECM (line 528-529)

8. Discussion: The authors mention studies like this could be used for novel drug target identification – could the authors elaborate this more with a sentence or two. A statement was added to elaborate on the fact that disease-modifying genes validated and functionalized using hiPSC models could be further developed as novel drug targets (lines 658-661).

9. Line 626: authors mention that NGN2 iNs have their limitations – what are these limitations? iNGN2 iNs provide a rapid method to obtain a synchronized population of excitatory neurons, and have great utility to detect pan-neuronal phenotypes that are not dependent on a very specific neuron subtype. However, the limitations are that NGN2 iNs have forced differentiation that may not fully reflect normal development, and that iN cultures have a less well-defined cell identity, as exhibited by previous reports describing the expression of peripheral nervous system markers. A statement describing this, along with relevant citations, has been added to the discussion (lines 673-681).

10. Discussion should include more discussion on the impact of the findings and potential translation into the clinic. A final paragraph has been added to discuss this point (lines 760-769).

---

## [Decision Letter · Decision Letter 1]

21 Aug 2023

Functional characterization of a single nucleotide polymorphism associated with Alzheimer’s Disease in a hiPSC-based neuron model.

PONE-D-23-13309R1

Dear Dr. Ramathal,

Thank you for addressing all the reviewer's comments. We’re pleased to inform you that your manuscript has been judged scientifically suitable for publication and will be formally accepted for publication once it meets all outstanding technical requirements.

Kind regards,

Amy McCart Reed

Academic Editor

PLOS ONE

Additional Editor Comments (optional):

Reviewers' comments:

Reviewer's Responses to Questions

**Comments to the Author**

1. If the authors have adequately addressed your comments raised in a previous round of review and you feel that this manuscript is now acceptable for publication, you may indicate that here to bypass the “Comments to the Author” section, enter your conflict of interest statement in the “Confidential to Editor” section, and submit your "Accept" recommendation.

Reviewer #1: All comments have been addressed

2. Is the manuscript technically sound, and do the data support the conclusions?

Reviewer #1: (No Response)

3. Has the statistical analysis been performed appropriately and rigorously? 

Reviewer #1: (No Response)

4. Have the authors made all data underlying the findings in their manuscript fully available?

Reviewer #1: (No Response)

5. Is the manuscript presented in an intelligible fashion and written in standard English?

Reviewer #1: (No Response)

6. Review Comments to the Author

Reviewer #1: (No Response)

7. PLOS authors have the option to publish the peer review history of their article (what does this mean?). If published, this will include your full peer review and any attached files.

Reviewer #1: No

---

## [Editor Report · Acceptance letter]

15 Sep 2023

PONE-D-23-13309R1 

Functional characterization of a single nucleotide polymorphism associated with Alzheimer’s Disease in a hiPSC-based neuron model. 

Dear Dr. Ramathal:

I'm pleased to inform you that your manuscript has been deemed suitable for publication in PLOS ONE. Congratulations! Your manuscript is now with our production department. 

Kind regards, 

on behalf of

Dr. Amy McCart Reed 

Academic Editor

PLOS ONE